# Programmed knockout mutation of liver fluke granulin attenuates virulence of infection-induced hepatobiliary morbidity

**Patpicha Arunsan[1,2,3†], Wannaporn Ittiprasert[2,3†], Michael J Smout[4†], Christina J Cochran[2,3], Victoria H Mann[2,3], Sujittra Chaiyadet[1], Shannon E Karinshak[2,3], Banchob Sripa[5], Neil David Young[6], Javier Sotillo[4], Alex Loukas[4‡*], Paul J Brindley[2,3‡*], Thewarach Laha[1†*]**

[1]Department of Parasitology, Faculty of Medicine, Khon Kaen University, Khon Kaen, Thailand; [2]Department of Microbiology, Immunology and Tropical Medicine, George Washington University, Washington DC, United States; [3]Research Center for Neglected Diseases of Poverty, School of Medicine & Health Sciences, George Washington University, Washington DC, United States; [4]Centre for Molecular Therapeutics, Australian Institute of Tropical Health and Medicine, James Cook University, Cairns, Australia; [5]Department of Pathology, Faculty of Medicine, Khon Kaen University, Khon Kaen, Thailand; [6]Faculty of Veterinary and Agricultural Sciences, The University of Melbourne, Victoria, Australia

**\*For correspondence:**
alex.loukas@jcu.edu.au (AL);
pbrindley@gwu.edu (PJB);
thewa_la@kku.ac.th (TL)

[†]These authors contributed equally to this work
[‡]These authors also contributed equally to this work

**Competing interests:** The authors declare that no competing interests exist.

**Abstract** Infection with the food-borne liver fluke *Opisthorchis viverrini* is the principal risk factor (IARC Working Group on the Evaluation of Carcinogenic Risks to Humans, 2012) for cholangiocarcinoma (CCA) in the Lower Mekong River Basin countries including Thailand, Lao PDR, Vietnam and Cambodia. We exploited this link to explore the role of the secreted growth factor termed liver fluke granulin (*Ov*-GRN-1) in pre-malignant lesions by undertaking programmed CRISPR/Cas9 knockout of the *Ov*-GRN-1 gene from the liver fluke genome. Deep sequencing of amplicon libraries from genomic DNA of gene-edited parasites revealed Cas9-catalyzed mutations within *Ov*-GRN-1. Gene editing resulted in rapid depletion of *Ov*-GRN-1 transcripts and the encoded *Ov*-GRN-1 protein. Gene-edited parasites colonized the biliary tract of hamsters and developed into adult flukes, but the infection resulted in reduced pathology as evidenced by attenuated biliary hyperplasia and fibrosis. Not only does this report pioneer programmed gene-editing in parasitic flatworms, but also the striking, clinically-relevant pathophysiological phenotype confirms the role for *Ov*-GRN-1 in virulence morbidity during opisthorchiasis.
DOI: https://doi.org/10.7554/eLife.41463.001

## Introduction

Liver fluke infection caused by species of *Opisthorchis* and *Clonorchis* remains a major public health problem in East Asia and Eastern Europe. *O. viverrini* is endemic in Thailand and Laos, where ~10 million people are infected with the parasite (*Sripa et al., 2011*). In liver fluke endemic regions, this infection causes hepatobiliary morbidity including cholangitis, choledocholithiasis (gall stones), and periductal fibrosis, and is the principal risk factor for bile duct cancer, cholangiocarcinoma (CCA) (*Sripa et al., 2011*; *Sripa et al., 2007*; *Mairiang et al., 2012*; *Tyson and El-Serag, 2011*; *Shin et al., 2010a*). Indeed, there is no stronger link between a human malignancy and a parasitic infection than that between CCA and infection with *O. viverrini* (*Pagano et al., 2004*). Northeastern Thailand suffers the highest incidence of CCA in the world, often exceeding 80 cases per 100,000 population and for which up to 20,000 people annually are admitted for surgery. The prognosis for liver fluke

**eLife digest** In the rural regions alongside the Mekong River in South East Asia, traditional cuisines often use uncooked or under cooked fish, many of which carry a worm known as *Opisthorchis viverrini*. Once inside the body, this parasite settles in the human liver, causing a tropical disease known as liver fluke infection. Out of the 10 million people affected by *O. viverrini*, thousands will also develop a type of liver cancer that is triggered by the presence of the worm. In particular, the parasite secretes a protein known as granulin that may encourage certain liver cells to multiply, potentially raising the risk for cancer.

A gene editing technique called CRISPR/Cas9 allows scientist to precisely target and then deactivate the genetic information a cell needs to produce a given protein. While the tool has been used in other species before, it was unknown if it could be applied to *O. viverrini*. Here, Arunsan et al. harnessed CRISPR/Cas9 to deactivate the gene that codes for granulin and create parasites that can only produce very little of the protein.

Hamsters infected with the gene-edited worms had fewer symptoms of liver fluke infection compared to those carrying normal *O. viverrini*. The animals with parasites that cannot produce granulin also had fewer changes to the liver that are associated with cancer. These findings confirm that granulin has a role in promoting liver fluke infection and liver cancer.

Alongside this work, *Ittiprasert et al.* used CRISPR/Cas9 to inactivate a gene in a species of worm that causes a human disease called schistosomiasis. Together, these findings demonstrate for the first time that the gene editing method can be adapted for use in parasitic worms, which are a major public health problem in tropical climates. This tool should help scientists understand how the parasites invade and damage our bodies, and provide new ideas for treatment and disease control.
DOI: https://doi.org/10.7554/eLife.41463.002

infection-induced cancer remains poor (*Sripa et al., 2011*; *Khuntikeo et al., 2015*; *Khuntikeo et al., 2016*; *Luvira et al., 2016*).

How and why opisthorchiasis induces cholangiocarcinogenesis is likely multi-factorial, including mechanical irritation of the biliary tract during migration and feeding of the liver fluke, secretion by the parasite of inflammatory molecules, and nitrosamines in fermented foods that are a dietary staple in northeastern provinces of Thailand (*Songserm et al., 2012*). To survive in the hostile host environment, parasitic helminths produce an assortment of excretory/secretory (ES) products including proteins with diverse roles at the host–parasite interface. This interaction has long been thought, but not fully understood, to modify cellular homeostasis and contribute to malignant transformation during chronic opisthorchiasis (*Brindley and Loukas, 2017*). Feeding activity of the liver fluke inflicts wounds in the biliary tree, resulting in lesions that undergo protracted cycles of repair and re-injury during chronic infection. The liver fluke secretes mediators that accelerate wound resolution in monolayers of cultured cholangiocytes, an outcome that is compromised following silencing of expression of the liver fluke secreted growth factor *Ov*-GRN-1 using RNA interference (*Papatpremsiri et al., 2015*; *Smout et al., 2015*). We hypothesize that proliferation of biliary epithelial cells induced by *Ov*-GRN-1 is a pivotal factor in maintenance and progression of a tumorigenic microenvironment in the liver during chronic opisthorchiasis.

Progress with development of genetic tools for functional genomic studies with platyhelminth parasites has been limited to date (*Hoffmann et al., 2014*). The use of clustered regularly interspaced short palindromic repeats (CRISPR) associated with Cas9, an RNA-guided DNA endonuclease, has revolutionized genome editing in biomedicine, agriculture and biology (*Hsu et al., 2014*; *Sander and Joung, 2014*). Progress with CRISPR/Cas9 in numerous eukaryotes including the nematodes *Caenorhabditis elegans*, *Strongyloides stercoralis* and *Strongyloides ratti* has been described (*Sander and Joung, 2014*; *Waaijers and Boxem, 2014*; *Lok et al., 2017*; *Gang et al., 2017*), but this form of gene editing has not been reported for flatworm parasites. Here, we deployed a CRISPR/Cas9-based approach, aiming to knockout (mutate) the *Ov*-GRN-1 gene and assess the virulence of gene-edited flukes *in vitro* and *in vivo* in a hamster model of opisthorchiasis.

## Results

### Programmed mutation of growth factor secreted by carcinogenic liver fluke

Following transfection of adult flukes with the gene-editing construct targeting *Ov*-GRN-1, the activity and efficiency of programmed editing was evaluated by two approaches. First, quantitative PCR (qPCR) was employed, which relies on the inefficiency of binding of a primer (here termed OVR-F) overlapping the target genomic sequence of the guide RNA (gRNA), that is where mutations are expected to have occurred, compared to the binding efficiency of flanking primers, that is outside the mutated region (flanking primers termed OUT-F and OUT-R) (*Figure 1A and B*). The ratio between the OVR-F and OUT-R products and OUT-F and OUT-R products provided an estimate of the amplification fold-reduction in the sample of CRISPR/Cas9-edited compared to genomic DNA (gDNA) from control, wild-type liver flukes at the target sequence of the sgRNA, that is the annealing site for the OVR primer (*Shah et al., 2015*; *Yu et al., 2014*). A reduction in relative fold amplification of 2.7% was detected in gDNA from the Cas9-treated worms (*Figure 1E*, *Figure 1—figure supplement 1C*). Second, to identify, quantify and characterize the mutations that arose in the genome of *Ov*-GRN-1-edited (termed Δ*Ov*-GRN-1) flukes, we used an amplicon-sequencing approach. A targeted (amplicon) sequence library was constructed from gDNA from some of the flukes (7 to 21 days after pCas-*Ov*-GRN-1 transfection). A fragment of 173 bp spanning the predicted site of the programmed double stranded break of *Ov*-GRN-1 was amplified from gDNA primed with oligonucleotides flanking 1496–1668 nt of *Ov*-GRN1. Adaptors and barcodes were ligated into the amplicon libraries. Deep sequencing of the amplicon libraries was undertaken using the Illumina MiSeq system. Insertion-deletion (INDEL)/mutation profiles in the sequence reads were compared in multiple sequence alignments with the reference template sequence, nt 1,496–1,668 of wild type *Ov*-GRN-1. The CRISPResso computational pipeline was used to quantify gene-editing outcomes and efficiency (*Canver et al., 2018*; *Pinello et al., 2016*); among >2 million reads aligned against the reference sequence, 27,640 sequence reads exhibited non-homologous end joining (NHEJ) mutations, including 170 reads with insertions (0.6%), 193 reads with deletions (0.7%) and 27,277 reads with substitutions (98.7%). Overall, 1.3% of the sequenced reads exhibited NHEJ mutations (*Figure 1C*). Regarding the NHEJ-bearing reads,>100 forms exhibited mutations that would disrupt the coding sequencing of *Ov*-GRN-1. Four representatives of the INDEL-bearing traces, aligned with the wild type (WT) allele are presented in *Figure 1—figure supplement 1B*. These and related (below) sequence reads are available at GenBank Bioproject PRJNA385864, Biosample SAMN07287348, SRA study SRP110673, accessions SRR5764463-5764618 and SRR8187484-SRR8187487, at https://www.ncbi.nlm.nih.gov/Traces/study/?acc=SRP110673, Bioproject, www.ncbi.nlm.nih.gov/bioproject/PRJNA385864.

### Diminished proliferation and wound healing induced by excretory/secretory products of genome-edited liver flukes

Effects of gene editing on transcription and protein expression in adult flukes were investigated. Levels of both *Ov*-GRN-1 mRNA transcripts as determined by reverse transcription (RT)-qPCR and of *Ov*-GRN-1 protein, as detected by western blot using anti-*Ov*-GRN-1 serum, fell significantly from days 1 and 2 after transfection, respectively (p≤0.0001; *Figure 1D and E*, *Figure 1—figure supplement 1C*). Expression levels of two reference genes encoding actin (*Figure 1—figure supplement 1C*) and the *Ov*-TSP-2 tegument protein (*Figure 1D*) were not influenced by the programmed mutation of *Ov*-GRN-1. These findings, revealing diminished RNA and protein following programmed mutation indicated that CRISPR/Cas9 catalyzed programmed gene-editing of *Ov*-GRN-1 was active in adult flukes *in vitro*. Thereafter, to investigate whether gene editing of *Ov*-GRN-1 impacted *in vitro* indicators of pathogenesis, the capacity of ES products from WT, mock-transfected and gene-edited flukes to drive proliferation and scratch wound repair of the H69 human cholangiocyte cell line was assessed. ES from WT and mock-transfected adult flukes stimulated cell proliferation and wound closure whereas an equivalent amount of ES products from Δ*Ov*-GRN-1 flukes resulted in significantly reduced cell proliferation over the 6-day course of the assay (p ≤ 0.0001; *Figure 2A and B*, *Figure 2—figure supplement 1A and B*) and significantly reduced *in vitro* wound closure over 36 hr

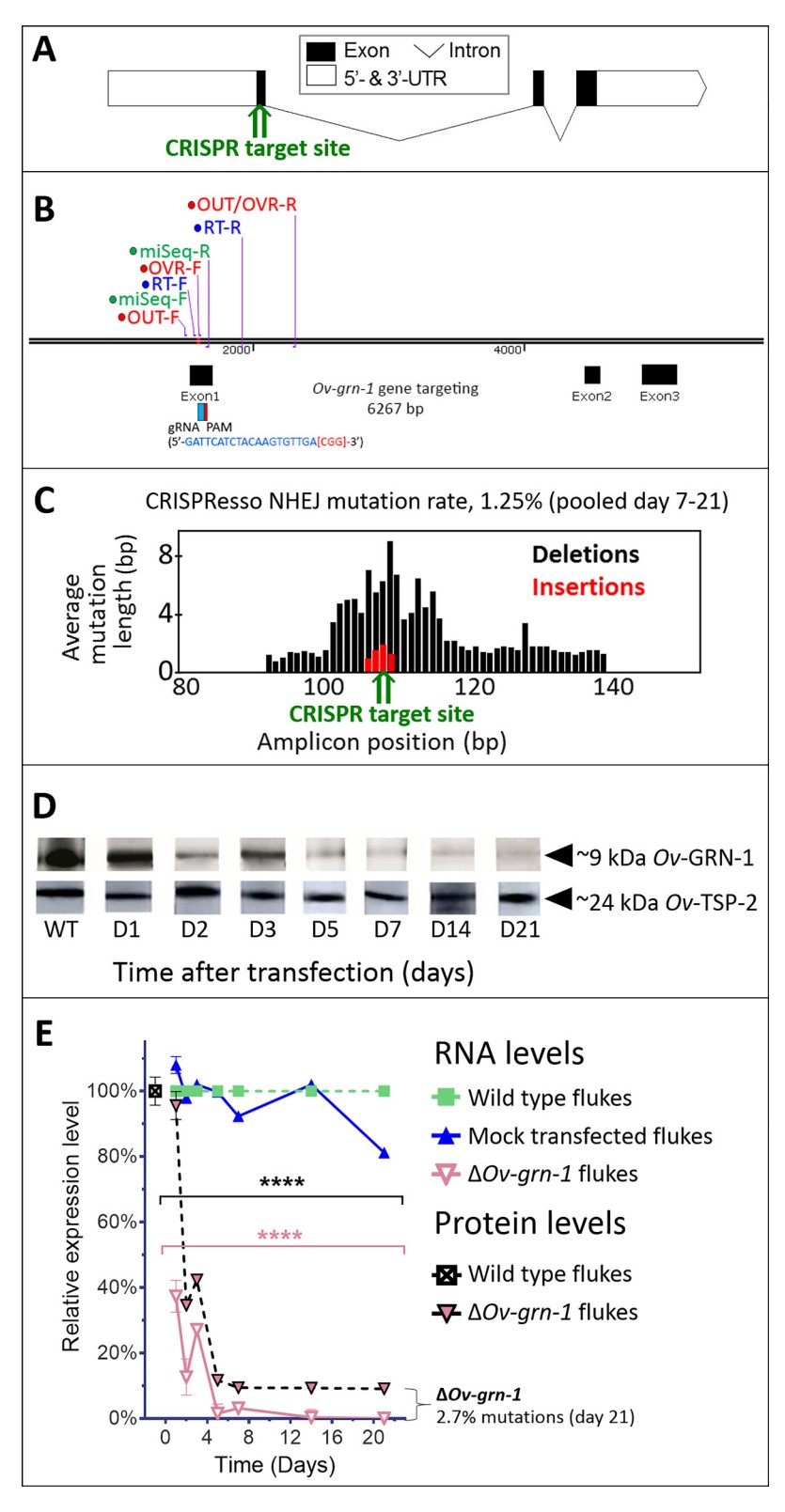

**Figure 1.** CRISPR/Cas9-mediated gene editing strategy to knockout *Ov*-GRN-1 in the adult developmental stage of the *O. viverrini* liver fluke. (**A**) Schematic depiction of *Ov*-GRN-1 gene with CRISPR/Cas9 target site in first exon marked with green arrow. (**B**) The exon1, 2 and 3 location of *Ov*-GRN-1 gene, size of 6267 bp and gRNA targeting exon 1: gRNA sequence is highlighted in blue and the PAM (CGG) in parenthesis in red. Primer pairs were used to detect levels of mRNA expression using a RT-qPCR assay (*Ov*-GRN-1 RT- forward or RT-F and *Ov*-GRN1 RT-reverse or RT-R), Tri-primers were

*Figure 1 continued*

used to detect the % relative fold amplicon or mutations (outside-forward or OUT-F, overlap-forward or OVR-F and reverse primer or OUT/OVR-R) and MiSeq forward and reverse (MiSeq-F and MiSeq-R) primers were used to prepare the NGS amplicon. (C) CRISPR/Cas9-catalyzed insertion (red bars) and deletion (black bars) mutations (INDELs) detected in the *Ov*-GRN-1 gene; target site of programmed CRISPR/Cas9 double strand break indicated by the green arrow. Average mutation length was plotted against *Ov*-GRN-1 gene amplicon position in base pairs (bp). (D) Somatic tissues of individual adult worms (in triplicate per time per group) were solubilized, electrophoresed in SDS-PAGE gels, transferred to nitrocellulose membrane and probed with anti-*Ov*-GRN-1 rabbit antibody. WT: wild-type control fluke tissues; D1 to 21: ΔOv-GRN-1 fluke tissues sampled the arrow highlighting the ~9 kDa *Ov*-GRN-1 band at increasing time points (days) following transfection and ΔOv-GRN-1 flukes showed similar levels of expression of *Ov*-TSP-2 protein (control antibody). D1 to 21, protein products from flukes days 1 to 21 following gene-editing treatment. Western blot strips probed with rabbit anti-*Ov*-TSP-2 antiserum, the arrow highlighting the band at ~24 kDa representing *Ov*-TSP-2. (E) Reduced levels of *Ov*-GRN-1 transcripts and *Ov*-GRN-1 protein after transfection of adult flukes with *Ov*-GRN-1 CRISPR/Cas9 construct using quantitative real-time PCR (mRNA) and densitometry of western blot signals (protein). Data were plotted relative to wild type (WT) fluke values (100%) as the mean ±SD of three replicates. ****$p < 0.0001$ compared to levels in WT flukes - protein in black; RNA in pink - at each time point (two-way ANOVA Holm-Sidak multiple comparison test).
DOI: https://doi.org/10.7554/eLife.41463.003

The following figure supplement is available for figure 1:

**Figure supplement 1.** CRISPR/Cas9 targeting *Ov*-GRN-1 design and fluke transfection.
DOI: https://doi.org/10.7554/eLife.41463.004

($p \leq 0.0001$; *Figure 2C and D*, *Figure 2—figure supplement 1C and D*), consistent with the reduction in *Ov*-GRN-1 protein secreted from the gene-edited liver flukes.

## Attenuated infection-induced hyperplasia of the biliary tract

Notwithstanding the marked effects observed with gene-edited, adult developmental forms, the metacercaria (MC) (*Figure 3A*) is the infective stage of *O. viverrini* for humans. Accordingly, we investigated gene knockout in MC. Significant differences in *Ov*-GRN-1 transcript levels were noted between groups of MC ($p \leq 0.01$), but the effect was modest, $\leq 4\%$, at each time point (*Figure 3—figure supplement 1*), suggesting that delivery of the pCas-*Ov*-GRN-1 by electroporation through the MC cyst wall was ineffective. Exposure to bile acids and gastric enzymes results in excystation of *O. viverrini* MC in the duodenum of the mammalian host (*Sripa et al., 2011*). Using trypsin, here the process was mimicked *in vitro* to release the newly excysted juvenile worms (NEJ) (*Figure 3B*), after which these NEJs were subjected to electroporation with the CRISPR/Cas9 plasmid construct, in like fashion to the adult developmental stage of *O. viverrini* (above). Following this manipulation, marked depletion of *Ov*-GRN-1 transcripts in NEJ was evident by 24 hr later ($p \leq 0.0001$) (*Figure 3C*).

In parallel, hamsters were infected with 100 ΔOv-GRN1 NEJs or WT NEJs immediately after electroporation. At necropsy of the hamsters 14 days later, similar numbers of WT and ΔOv-GRN-1 flukes were observed in the bile ducts, and they were similarly motile (not shown). Strikingly, however, acute infection with ΔOv-GRN-1 parasites failed to induce the marked hyperplasia of the biliary epithelia characteristic of chronic opisthorchiasis. Specifically, infection with WT flukes induced markedly disordered, hyperplasic growth of the epithelium adjacent to the parasites; ~500% thickening of the biliary epithelium compared to uninfected controls as measured in two-dimensional image analysis of H and E-stained thin sections ($p \leq 0.0001$). By contrast, infection with the ΔOv-GRN-1 flukes provoked significantly less ($p \leq 0.0001$) biliary hyperplasia than WT flukes (145% thickening compared to uninfected controls; $p \leq 0.01$). Indeed, the bile ducts from hamsters infected with the ΔOv-GRN-1 flukes generally resembled those of the uninfected control hamsters (*Figure 3D–G*). At 60 days after infection, significant differences in biliary hyperplasia remained between hamsters infected with WT (216%) and ΔOv-GRN-1 (162%) flukes ($p \leq 0.05$), although this was less marked than during acute infection at day 14 (*Figure 3G*).

## Reduced periductal fibrosis and morbidity during chronic opisthorchiasis

To evaluate disease during chronic infection with ΔOv-GRN-1 liver flukes and associated chronic biliary morbidity, hamsters were infected with ΔOv-GRN-1 and WT NEJ, and adult flukes were recovered and counted from the livers 60 days post-infection. Similar numbers of worms were recovered from both control and gene-edited liver fluke-infected hamsters (*Figure 4A*). To assess the impact of infection with ΔOv-GRN-1 on markers of chronic opisthorchiasis including biliary fibrosis, liver

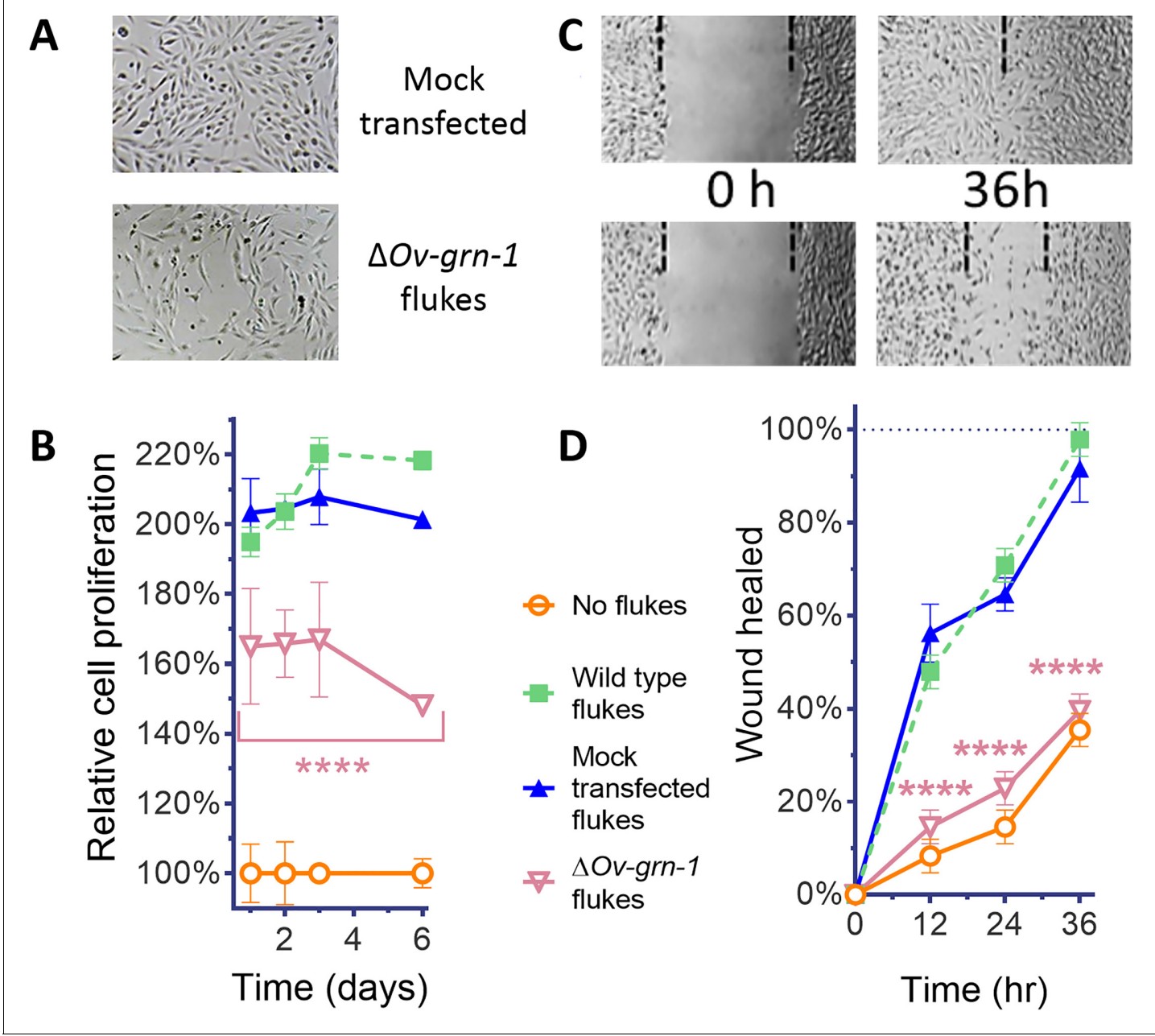

**Figure 2.** ES products of Δ*Ov*-GRN-1 adult flukes induced less cell proliferation and wound repair *in vitro*. (A) Representative cell proliferation images of H69 cholangiocyte cells co-cultured with flukes in Transwell plates; mock transfected (top) and Δ*Ov*-GRN-1 (bottom) groups shown at day 3. (B) Reduced cell proliferation induced by ES products of Δ*Ov*-GRN-1 fluke, as shown in panel (A), quantified from days 1 to 6. Data were plotted as mean relative percentage to control cells cultured in the absence of flukes. (C) Representative image of repair of wound from scratch in cultured H69 scratch during co-culture with flukes in Transwell plates. Mock transfected (upper) and Δ*Ov*-GRN-1 (lower) groups shown at 0 and 36 hr after scratch wounding. Dotted line indicates the margin of the wound. (D) Scratch wound repair assay quantified from 0 to 36 hr, revealing diminished healing in cells co-cultured with the Δ*Ov*-GRN-1 parasites. Panels (B) and (D): mean ±SD, three replicates; ****p < 0.0001 compared to wild-type flukes with two-way ANOVA Holm-Sidak multiple comparison test.

DOI: https://doi.org/10.7554/eLife.41463.005

The following figure supplement is available for figure 2:

**Figure supplement 1.** Extended set of images showing Δ*Ov*-GRN-1 adult fluke ES products induce less *in vitro* cell proliferation and wound repair.

DOI: https://doi.org/10.7554/eLife.41463.006

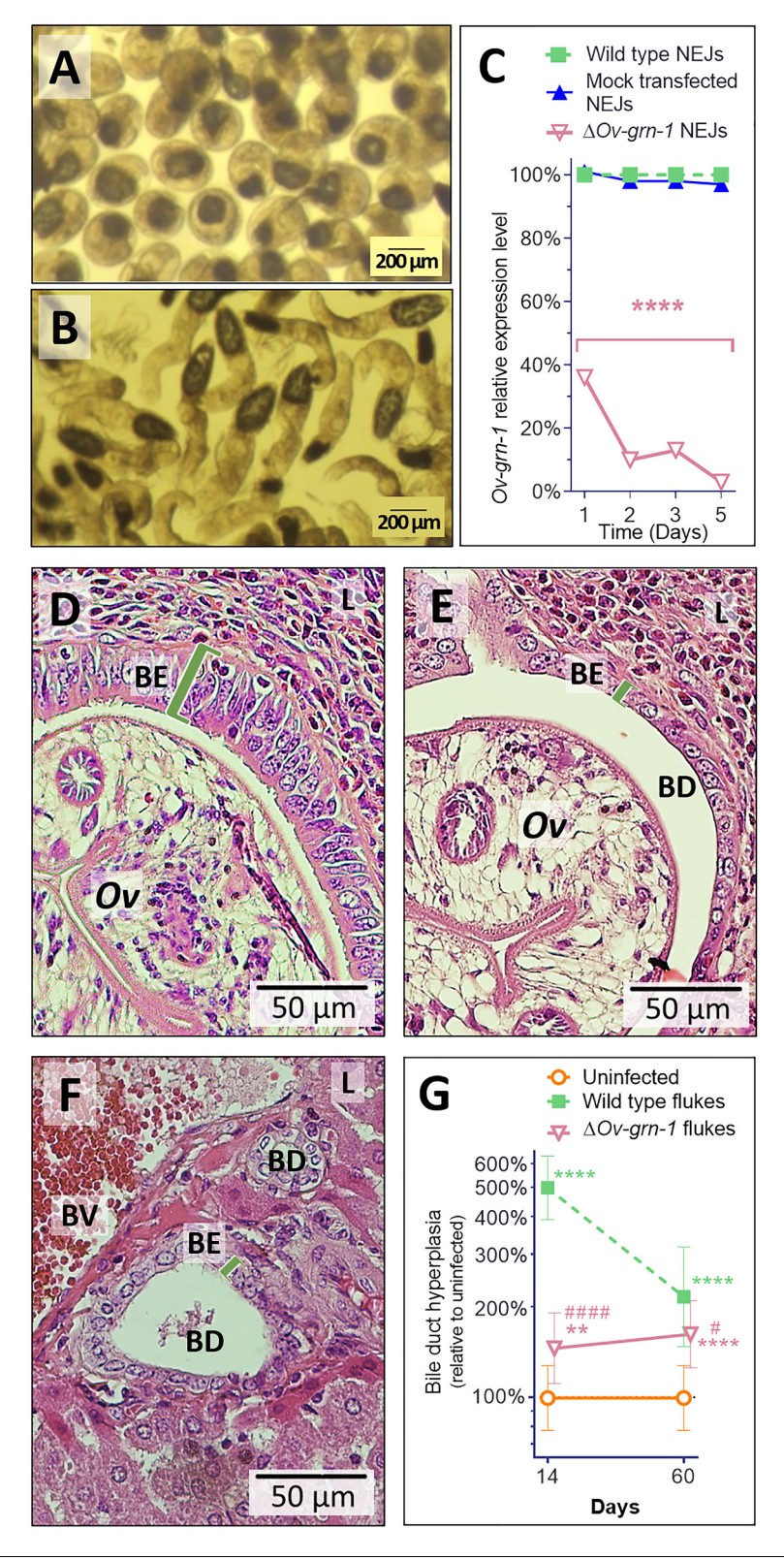

**Figure 3.** Gene edited ΔOv-GRN-1 newly excysted juveniles infected hamsters but drove reduced acute pathogenic lesions. Micrographs of metacercariae (**A**) and newly excysted juvenile flukes (NEJ) (**B**). (**C**) Levels of Ov-GRN-1 transcripts in mock-transfected and ΔOv-GRN-1 flukes at one to five days as quantified with qPCR and plotted relative to the wild type (WT) untreated group; mean ±SD; three replicates; ****p ≤ 0.0001 compared to WT flukes with two-way ANOVA Holm-Sidak multiple comparison test at each time point. (**D**) Representative micrograph (200 × magnification) of H&E-
*Figure 3 continued on next page*

*Figure 3 continued*

stained thin sections from livers of hamsters at 14 days after infection with WT flukes. (**E**) Representative micrograph of H&E-stained thin sections showing hamster liver 14 days after infection with ΔOv-GRN-1 flukes. (**F**) Representative micrograph of H&E stained thin sections of livers of control, uninfected hamsters, revealing the healthy, organized pavement-like profile of the cells of the biliary epithelium (BE) enclosing the lumen of the bile duct (BD), near a blood vessel (BV), within the liver (L). Infection by WT flukes (**D**) revealed thickened, disordered epithelium adjacent to the parasite (Ov). Infection with the gene edited ΔOv-GRN-1 flukes (**E**) revealed a bile duct epithelium more similar to the uninfected hamster. (**G**) Epithelium width/hyperplasia (green bracket) was quantified using ImageJ and plotted as the mean ±SD of five biological replicates (hamsters) from each of group and time point (14 and 60 days). Significant differences were apparent when compared to the uninfected group using the two-way ANOVA with Holm-Sidak multiple comparison test: **$p \leq 0.01$ and ****$p \leq 0.0001$, and wild-type compared to ΔOv-GRN-1, #$p \leq 0.05$ and ####$p \leq 0.0001$.
DOI: https://doi.org/10.7554/eLife.41463.007

The following figure supplement is available for figure 3:

**Figure supplement 1.** CRISPR/Cas9 ineffective at silencing gene expression in metacercariae (MC).
DOI: https://doi.org/10.7554/eLife.41463.008

sections from infected hamsters were stained with Picro-Sirius Red to localize collagen bundles in the biliary tract (*Figure 4B*). Minimal deposits of collagen were seen in the periductal regions of the biliary tract of the uninfected control hamsters. By contrast, thick bands of collagen surrounded the enlarged bile ducts in the vicinity of the flukes in the hamsters infected with WT parasites. Significantly less collagen (28%) had been deposited in periductal regions of hamsters infected with ΔOv-GRN-1 flukes compared to livers of hamsters infected with WT flukes ($p \leq 0.001$) (*Figure 4B and C*). To further assess fibrosis, thin sections of livers were immuno-stained for alpha-smooth muscle actin (α-SMA or ACTA2), a marker of hepatic fibrosis (*Guido et al., 1997*). Livers of hamsters infected with WT flukes showed densely packed collagen fibrils that stained for ACTA2 in periductal regions proximal to the parasites. In contrast, livers from hamsters infected with ΔOv-GRN-1 flukes displayed an irregular distribution of less dense collagen fibrils with less ACTA2-specific fluorescence (*Figure 4D*, *Figure 4—figure supplement 1*). Measuring Alexa-594 fluorescence quantified the expression levels of ACTA2. Median levels of ACTA2 (quantified using Alexa-594-anti-ACTA2) in the livers of ΔOv-GRN-1 fluke-infected hamsters were significantly reduced (94%) compared to those of WT fluke-infected hamsters ($p \leq 0.01$) (*Figure 4E*).

## Gene editing efficiency correlated negatively with granulin gene expression

Bile ducts parasitized by the gene-edited worms displayed a broad range of fibrosis from minimal to marked, as established by staining both with Sirius Red and with antibody specific for alpha-smooth muscle actin. This situation may have reflected unevenness in level of programmed mutation of the Ov-GRN-1 gene in cells within and/or among individual liver flukes. To investigate this situation further, we assessed transcription of the Ov-GRN-1 gene from individual adult flukes recovered from hamsters 60 days after infection with gene edited NEJ. This revealed that levels of Ov-GRN-1 mRNA in the ΔOv-GRN-1 group flukes were 81% lower, in aggregate, than the control wild-type flukes (*Figure 5A*). Thereafter, to evaluate the mutation rate of the gene editing approach, which involved transfection by electroporation of batches of 750 NEJs, adult flukes at necropsy were assigned to one of three groups based on Ov-GRN-1 mRNA expression levels, as follows: (i) ≥ 100% relative to WT mean, that is, low (L) efficiency of programmed gene editing; group was termed $_L$ΔOv-GRN-1; (ii) > 10 to<100% relative to WT mean, that is moderate (M) level efficiency of programmed gene editing; termed $_M$ΔOv-GRN-1; and (iii) ≤ 10% relative to WT mean, that is high (H) level efficiency of programmed gene editing; termed $_H$ΔOv-GRN-1. Genomic DNAs pooled from 7 to 10 worms of each group were studied to quantify the efficiency of gene editing, using both the NGS CRISPResso and the tri-primer qPCR approaches. The NGS CRISPResso analysis revealed mutation rates of 1.3, 5.9 and 17.2% in the L, M, and H groups of ΔOv-GRN-1 worms, respectively. The tri-primer qPCR analysis indicated mutation levels of 0.7, 3.2 and 4.6% in these groups, respectively. Both approaches confirmed that the efficiency of programmed gene editing negatively correlated with levels of the Ov-GRN-1 transcripts (*Figure 5A and B*). The combined mutation frequency among all three groups by the two approaches was 8.1% and 2.7%, with the 2.7% rate estimated by tri-primer qPCR indicating the same level as the mutation rate of 2.7% observed during culture of adult stage ΔOv-GRN-1 flukes for 7 to 21 days *in vitro* (*Figure 1E*, *Figure 1—figure supplement 1C*). In

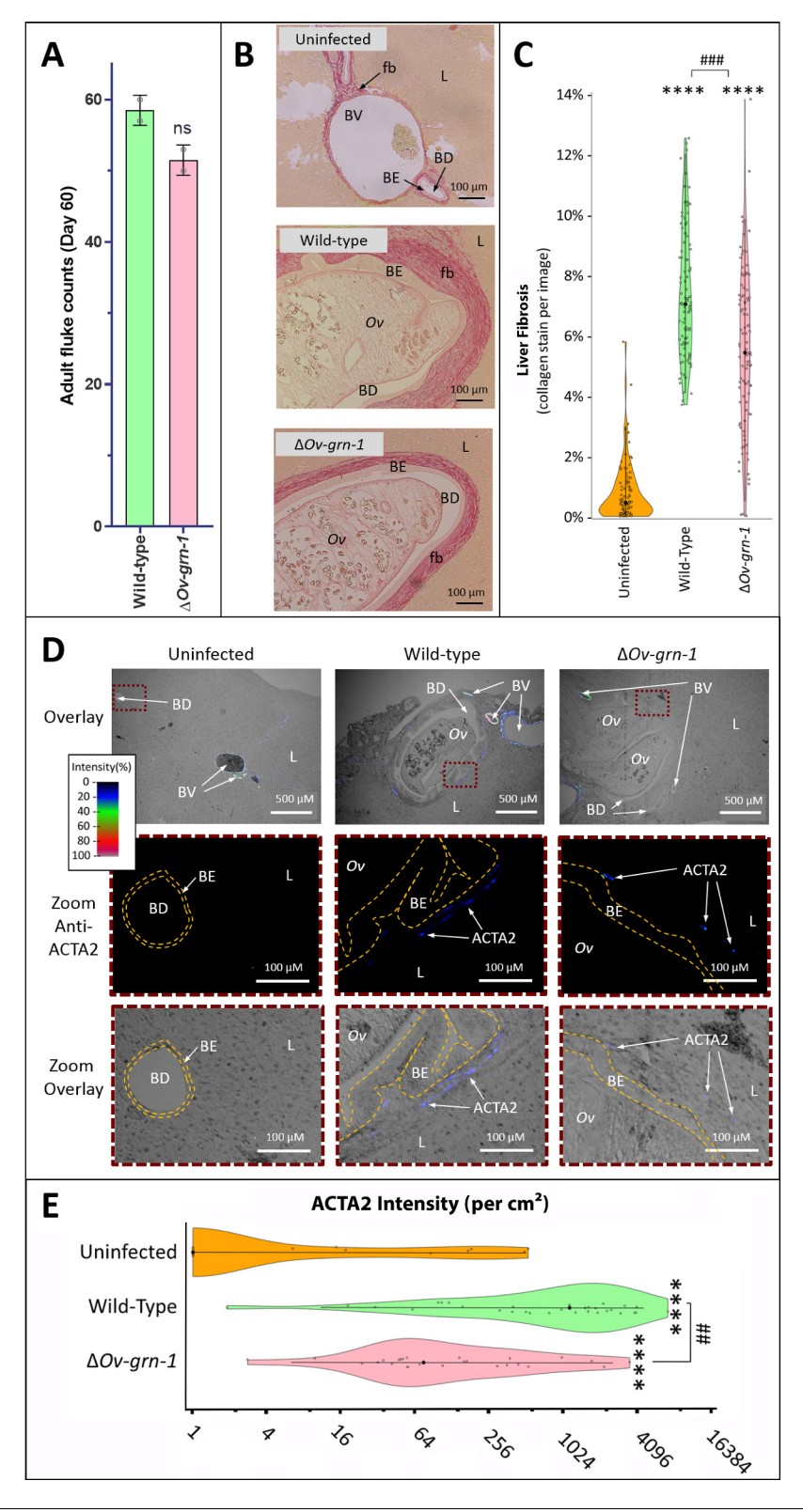

**Figure 4.** Reduced fibrosis during chronic infection of hamsters with gene-edited Δ*Ov*-GRN-1 liver flukes. (**A**) Adult fluke numbers were counted from the livers of necropsy at 60 days post-infection and presented as the average and range for two hamsters per group. Numbers of flukes were similar in the wild type (WT) and Δ*Ov*-GRN-1 groups; non-significant (ns). (**B**) Representative micrographs of thin sections of Sirius red-stained liver (**L**) from uninfected hamsters revealed minimal deposition of collagen (fibrosis [fb]) in the periductal regions and adjacent liver parenchyma; a thin margin of

*Figure 4 continued on next page*

*Figure 4 continued*

red-stained material outlined the endothelial cells of the blood vessel (BV) walls, and the biliary epithelia (BE) of the bile ducts (BD). Livers from hamsters infected with WT flukes (*Ov*) included marked deposition of collagen with elongated BE cells adjacent to the flukes. There was substantial collagen deposition in livers of hamsters infected with Δ*Ov*-GRN1 flukes compared to uninfected liver sections but far less than for hamsters infected with WT flukes. (C) Liver fibrosis quantified with ImageJ MRI-fibrosis plugin presented as violin plots: 100 images containing bile ducts from 20 sections (five hamsters) per group; mean (black dot)±SD (vertical line). Fibrosis was reduced in the Δ*Ov*-GRN-1 (23% less) compared to WT fluke-infected hamsters. The width of the violin plot represents measurement frequency. The Kruskal-Wallis with Dunn's multiple comparisons test was used to compare groups against the uninfected hamsters: ****p $\leq$ 0.0001; and Δ*Ov*-GRN-1 against WT, ###p $\leq$ 0.001. (D) Representative micrographs, immunofluorescence/bright-field overlays, of sections probed with anti-ACTA2 antibody with fluorescence intensity indicated on a blue/green/red scale. ACTA2 was universally detected in myofibroblasts surrounding BV but not detected adjacent to healthy uninfected BD. The proximity of ACTA2 to fluke-infected BD was suggestive of myofibroblast generation in response to fluke-induced damage to BE. The upper row of micrographs (overlay) present combined bright-field and anti-ACTA2 fluorescence wide views of the liver sections. The boxed regions in the upper row indicate informative sites, which have been magnified and expanded in the central and lower rows of micrographs. The central row presents the boxed region with anti-ACTA2 fluorescence alone (Zoom Anti-ACTA2) and lower row presents the bright-field image overlaid by the fluorescence field (Zoom Overlay). Liver sections exhibited intense fluorescence surrounding BV (arterial blood vessels: red/green, venous vessels: blue/green), whereas in livers of uninfected hamsters BD exhibited only minimal fluorescence. The highlighted magnified (Zoom) rows of images revealed WT infected livers expressing mild (blue) but steady levels of ACTA2-staining surrounding thickened BE layer. The inner and outer BE cell margins are indicated by the dotted line (orange) around BDs with WT flukes. Livers from hamsters infected with Δ*Ov*-GRN1 flukes showed irregular, generally feeble expression of ACTA2 proximal to BD. (E) Quantified levels of ACTA2 signals surrounding BDs from sections of hamster livers. Violin plot with reverse log2 Y-axis showing the ACTA2 intensity (per cm$^2$ at 300 PPI) adjacent to BE, established from 25 to 30 discrete BD images per group (three hamsters), as assessed with ImageJ. Zero values from the uninfected group were deemed to have a value of 1 in order to plot the log axis. SD indicated as a line with the mean indicated by the central black dot, and the width of the violin indicative of frequency of measurement. ACTA2 staining showed 94% median reduction in Δ*Ov*-GRN-1 fluke-infected livers compared to hamsters infected with WT liver flukes. One-way ANOVA with Holm-Sidak multiple comparison test, ****p $\leq$ 0.0001 compared to uninfected and ##p < 0.01 compared to Δ*Ov*-GRN1 flukes against WT flukes.

DOI: https://doi.org/10.7554/eLife.41463.009

The following figure supplement is available for figure 4:

**Figure supplement 1.** Representative wide-angle view of anti-ACTA2 immunofluorescence and bright field liver sections.
DOI: https://doi.org/10.7554/eLife.41463.010

addition, the NGS CRISPResso analysis of the sequence reads of the gene-edited L, M and H groups compared with those from the control WT group (GenBank accessions SRR8187484-SRR8187487, 5 to 10 million reads per targeted amplicon library) provided details of the nature and types of the mutations as insertions, deletions and/or substitutions following NHEJ events that repaired the programmed cleavage of the *Ov*-GRN-1 locus. The analysis also revealed increasing ratio of substitutions among the mutations among the $_L$Δ*Ov*-GRN-1, $_M$Δ*Ov*-GRN-1 and $_H$Δ*Ov*-GRN-1 groups (*Figure 5B*). Lastly, these findings also demonstrated the longevity of the programmed mutation at *Ov*-GRN-1; mutations were retained in the parasite for at least 60 days during active infection of the mammalian host.

## Discussion

This report, and the accompanying article on schistosomes (*Ittiprasert et al., 2019*), pioneer programmed gene editing using CRISPR/Cas9 of trematodes and indeed genome editing for species of the phylum Platyhelminthes. The findings revealed that somatic tissue gene editing disrupted the expression of liver fluke granulin, resulting in a clinically noteworthy phenotype of attenuated hepatobiliary tract morbidity. Scrutiny of the nucleotide sequence reads indicated that the chromosomal break took place as programmed and was repaired subsequently by NHEJ following Cas9-catalyzed mutation (*Albadri et al., 2017*). Accordingly, the findings confirmed that the bacterial Type II Cas9 system is active in *O. viverrini*, and we suggest that Cas9-mediated programmed gene editing and repair by homology directed repair and NHEJ will be active in other genes of the liver fluke, and in other trematodes and parasitic platyhelminths generally.

Although the findings demonstrated programmed gene editing of the *Ov*-GRN-1 locus, the somatic mutation rate in the adult developmental stage was generally <5% of the genomes recovered from these multicellular parasites. This low mutation rate contrasted with both the marked reduction in *Ov*-GRN-1 message detected *in vitro* and the pathophysiological outcomes and reduced virulence of infection of hamsters with gene-edited flukes. The anomaly might be explained

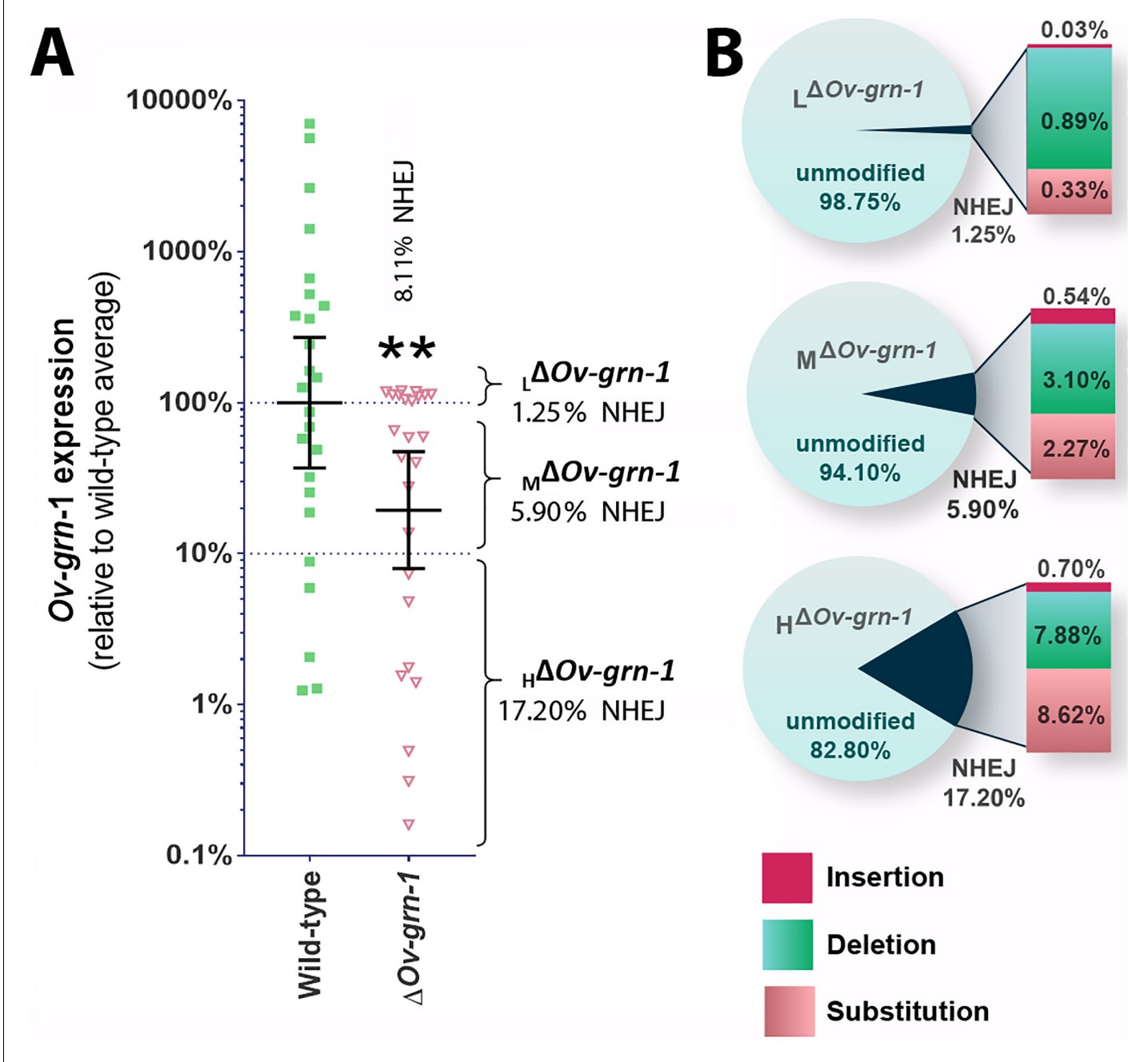

**Figure 5.** Characterization of gene knockout mutations in adult liver flukes from hamsters chronically infected hamsters. Expression of *Ov*-GRN-1 transcripts was reduced in Δ*Ov*-GRN-1 compared to WT flukes. (A) Reverse $\log_{10}$ Y-axis shows the qPCR $2^{(-\Delta\Delta Ct)}$ findings from flukes 60 days after programmed CRISPR/Cas9-gene editing and hamster infection plotted relative to mean value for the WT infection. The WT group displayed a broad level of expression, whereas the mean expression level for the Δ*Ov*-GRN-1 flukes was only 19.4% of the WT group; Mann-Whitney nonparametric test, **$p \leq 0.01$. Although significantly lower as a group, individual worms of this Δ*Ov*-GRN-1 cohort displayed phenotypes that ranged from no apparent effect (wild-type phenotype) to markedly diminished expression of *Ov*-GRN-1 (Δ*Ov*-GRN-1 phenotype). Mutation frequency was assessed by assigning worms in the Δ*Ov*-GRN-1 group into three sub-groups of flukes based on CRISPR/Cas9 mutation frequency. Eight flukes with effective CRISPR gene knockout (H Δ*Ov*-GRN1: *Ov*-GRN-1≤10% expression), seven flukes with modest levels of transcript knockout (M Δ*Ov*-GRN-1:>10 to<100% *Ov*-GRN-1), and 10 flukes exhibiting little or no effect (L Δ*Ov*-GRN-1: *Ov*-GRN-1 100–120%). The aggregate mutation frequency among the three Δ*Ov*-GRN-1 sub-groups was 8.1% and 2.7% as estimated by the NGS-CRISPResso and tri-primer qPCR approaches, respectively. (B) Pie charts showing the CRISPResso estimated rate of programmed mutations in the L Δ*Ov*-GRN-1, M Δ*Ov*-GRN-1 and H Δ*Ov*-GRN1 sub-groups compared to non-mutated, WT reads. NHEJ mutation rate indicated by the pie slice (navy blue) and the proportion of each mutation sub-type is provided on the expanded bar at the right.
DOI: https://doi.org/10.7554/eLife.41463.011

by the tissue expression of secreted *Ov*-GRN-1. Although it exhibits generalized expression throughout tissues of the adult liver fluke, predominant expression of *Ov*-GRN-1 has been immunolocalized to the tegumental surface, tegumental cytons and gut (*Smout et al., 2009*). Given that the flukes were transfected *in vitro* with the gene editing plasmid by square wave electroporation, gene knockout of the target *Ov*-GRN-1 locus in nuclei of cells in the tegument and gut may have occurred more frequently than in cells deeper within the fluke. If so, this may explain the marked reduction of expression and secretion of *Ov*-GRN-1 in tandem with a limited rate of mutation estimated in genomic DNA pooled from the gene-edited flukes.

The activity *in vitro* of liver fluke granulin in cell proliferation, wound repair and angiogenesis has been established (*Papatpremsiri et al., 2015*; *Smout et al., 2015*; *Smout et al., 2009*), which has prompted the development of therapeutic peptides based on the *Ov*-GRN-1 scaffold for treatment of non-healing wounds (*Bansal et al., 2017*; *Dastpeyman et al., 2018*). The novel findings reported here corroborate earlier *in vitro* reports and extend the findings in a rodent model of human opisthorchiasis. Programmed gene editing confirmed that secreted parasite granulin induces hyperplasia of the biliary epithelium and fibrosis during chronic infection, and that liver fluke granulin directly contributes to morbidity of the hepatobiliary tract during both acute and chronic opisthorchiasis. The impact of *Ov*-GRN-1 might emulate the action of interleukin IL−33, an epithelial mitogen for cholangiocytes, in the development of CCA. IL-33 primes type two innate lymphoid cells to induce proliferation of neighboring cholangiocytes by the release of IL-13 (*Brindley and Loukas, 2017*; *Li et al., 2014*). The pathophysiological bioactivity of granulin warrants deeper investigation of its role in fibrosis, including the influence on hepatic stellate cells, during liver fluke infection and cholangiocarcinogenesis (*Guido et al., 1997*; *Yin et al., 2013*; *Gouveia et al., 2017*; *Rockey et al., 2015*).

The rigor of future gene editing investigations might be enhanced with the inclusion of additional controls including parasites transfected with an otherwise functional vector that lacks target-specific gRNA and/or a gRNA with a scaffold but without seed sequence and/or containing a seed sequence without homology in the genome of *O. viverrini*. These additional controls would address non-target-specific effects of expression of Cas9 including on the genetic fitness of the genome-edited parasites (*Cox et al., 2015*; *Kosicki et al., 2018*; *Ihry et al., 2018*). Likewise, in addition to estimation of gene-editing performance and efficiency of somatic cell gene-editing in this multicellular helminth parasite using NGS-based (*Shah et al., 2015*; *Canver et al., 2018*; *Albadri et al., 2017*) and quantitative PCR-based approaches (*Shah et al., 2015*; *Yu et al., 2014*), droplet digital PCR (ddPCR)-based analysis should provide more sensitive detection and quantification of gene-editing manipulations. The ddPCR approach can provide simultaneous assessment of both homology directed repair and NHEJ, the repair pathways that resolve Cas9 catalyzed double-stranded breaks, and also investigate multiple, simultaneous editing conditions at the target locus (*Miyaoka et al., 2018*). With respect to *Ov*-GRN1 and its tissue site of expression, the anomaly between the marked knockdown of transcript levels and the minority of genomes mutated by the programmed gene editing among the total number of cells in this liver fluke, is amenable to deeper inquiry. Characterizing by immunolocalization the site of expression in the parasite from hamsters infected with gene-edited NEJ and/or the location of the gene editing plasmid after transfection of the liver fluke should be instructive.

The causative agent for many cancers remains obscure including non-liver fluke infection-associated CCA. By contrast, the principal risk factor in liver fluke-endemic regions is well established: infection with *O. viverrini* and related parasites (*IARC Working Group on the Evaluation of Carcinogenic Risks to Humans, 2012*; *Fedorova et al., 2017*; *Shin et al., 2010b*). CRISPR/Cas9-based gene editing and the hamster model of human opisthorchiasis utilized here (*Sripa et al., 2007*), including genetic manipulation of the larval infective stage of the parasite, provide a facile, functional genomics system to interrogate this host-parasite relationship and pathophysiology (*Hoffmann et al., 2014*). In a related model, periductal fibrosis induced by the liver fluke infection combined with ingestion of dimethylnitrosamine or similar nitric oxide carcinogen results in epithelial hyperplasia, cholangiocyte proliferation and DNA damage, which culminates in CCA (*Thamavit et al., 1987*; *Maksimova et al., 2017*). Investigation utilizing genome edited liver flukes, mutated at loci encoding granulin or other parasite products can now proceed, including interaction of liver fluke granulin with cholangiocyte signaling pathways that are frequently mutated during liver fluke infection-induced CCA (*Jusakul et al., 2017*).

# Materials and methods

## Key resources table

| Reagent type (species) or resource | Designation | Source or reference | Identifiers | Additional information |
|---|---|---|---|---|
| Hamster | *Mesocricetus auratus*, Syrian golden | Mahidol University, Thailand | NA | |
| Parasite (liver fluke) | *Opisthorchis viverrini*, adult development stage | (*Papatpremsiri et al., 2015*; *Laha et al., 2007*; *Sithithaworn et al., 1997*; *Pinlaor et al., 2013*) | NA | Infective stage metacercariae of *O. viverrini* obtained from naturally infected, cyprinid freshwater fish from Mukdahan and other Isaan provinces, Thailand, and used to infect hamsters. |
| Parasite (liver fluke) | *Opisthorchis viverrini*, metacercariae and newly excysted juveniles | (*Papatpremsiri et al., 2015*; *Laha et al., 2007*; *Sithithaworn et al., 1997*; *Pinlaor et al., 2013*; *Papatpremsiri et al., 2016*) | NA | Infective stage metacercariae of *O. viverrini* obtained from naturally infected, cyprinid freshwater fish from Isaan provinces, Thailand. |
| Commercial kit | GeneArt CRISPR Nuclease Vector Kit | Invitrogen | A21174 | |
| Serum | Anti-*Ov*-GRN-1 rabbit serum | (*Smout et al., 2009*) | NA | |
| Antibody | Alexa Fluor 594-labeled anti-ACTA2 | Abcam (*Capone et al., 2016*; *Burwinkel et al., 2018*) | ab202368 | |
| Commercial reagent | Agencourt AMPure XP beads | Beckman | A63880 | |
| Commercial reagent | GeneRead Adaptors I Set A | Qiagen | 180985 | |
| Commercial kit | QIAseq 1-step Amplicon Library Kit | Qiagen | 180412 | |
| Commercial kit | GeneRead Library Quant Kit | Qiagen | 180612 | |
| Human cholangiocyte cell line, H69 | Cholangiocyte cell line derived by SV 40 transformation | (*Smout et al., 2015*; *Grubman et al., 1994*) | H69; RRID: CVCL_8121 | |
| Commercial kit | Picro Sirius Red Stain Kit (Connective Tissue Stain) | Abcam | ab150681 | |

## *Opisthorchis viverrini* liver flukes

Metacercariae (MC) of *O. viverrini* were isolated from the naturally infected cyprinid fish by pepsin digestion as described (*Pinlaor et al., 2013*). In brief, fishes were homogenized using an electric blender, after which the homogenate was incubated for 120 min at 37°C in 0.25% porcine pepsin, 1.5% HCl, 150mM NaCl. Subsequently, the digest was filtered sequentially through sieves of 1100, 350, 250 and 140 µm diameter pore size. After gravity sedimentation of the final filtrate, the aqueous supernatant was discarded, the MC-enriched sediment was washed once in 150 mM NaCl, and the identity of MC as *O. viverrini* confirmed using a stereomicroscope. Batches of MC were stored in 150 mM NaCl at 4°C. The newly excysted-juvenile flukes (NEJ) were liberated from MC by incubation in 0.25% trypsin in $1\times$ PBS supplemented with $2\times$ 200 U/ml penicillin, 200 µg/ml streptomycin (Gibco) ($2\times$ Pen/Strep) for 5 min at 37°C in 5% $CO_2$ atmosphere, after which NEJ were separated from the discarded cyst walls of the MC by mechanical passage through a 27G (insulin) needle (*Papatpremsiri et al., 2015*; *Papatpremsiri et al., 2016*). Before use, NEJ were transferred into

RPMI medium supplemented with 1% glucose, 2 g/l NaHCO$_3$, 2× Pen/Strep, 1µM E-64 (Thermo Fisher Scientific) for 60 min at 37°C in 5% CO$_2$ atmosphere.

To obtain the adult developmental stage of the liver fluke, Syrian golden hamsters (*Mesocricetus auratus*) were infected by intragastric tube at 6–8 weeks of age with 50 MC per hamster (*Sripa and Kaewkes, 2002*). The hamsters were maintained at the rodent facility of the Faculty of Medicine, Khon Kaen University, Khon Kaen. Sixty days after infection, hamsters were euthanized, and the liver flukes collected as described (*Sripa and Kaewkes, 2002*). The Animal Ethics Committee of Khon Kaen University approved the study, approval number ACUC-KKU-61/60, which adhered to standard guidelines of the National Research Council of Thailand for the Ethics of Animal Experimentation.

## Vector and guide RNA targeting exon 1 of *Ov*-GRN-1

To edit the gene Ov-GRN-1 that encodes *O. viverrini* granulin-1 (6,287 bp, mRNA GenBank FJ436341.1) (*Smout et al., 2009*; *Young et al., 2014*), online tools including CRISPR design, http://crispr.mit.edu/ (*Ran et al., 2013*) and ChopChop, http://chopchop.cbu.uib.no/ (*Labun et al., 2016*; *Montague et al., 2014*) were employed to design a single guide RNA (sgRNA) targeting exon 1 of the *Ov*-GRN-1 gene at nucleotide position 1589–1608, 5'-GATTCATCTACAAGTGTTGA (*Figure 1A and B*). The programmed cleavage site was predicted to be located at three residues upstream of a CGG proto-spacer adjacent motif (PAM) sequence in exon 1 of Ov-GRN-1 (*Figure 1B*, *Figure 1— figure supplement 1B*). A CRISPR/Cas9-encoding vector encoding this sgRNA under the control of the mammalian U6 promoter and encoding Cas9 (with nuclear localization signal 1 and 2) driven by the CMV promoter was assembled (GeneArt CRISPR Nuclease Vector Kit, Invitrogen), and termed pCas-*Ov*-GRN-1 (*Figure 1—figure supplement 1A*). *Escherichia coli* TOP-10 competent cells were transformed with pCas-*Ov*-GRN-1 after which the plasmid was recovered from cultures of a positive clone (NucleoBond Xtra Midi, Macherey-Nagel GmbH, Germany). The nucleotide sequence of pCas-*Ov*-GRN-1 was confirmed by Sanger direct cycle sequencing.

## Transfection of liver flukes with pCas-*Ov*-GRN-1

Pools of 20 mature adult flukes were simultaneously subjected to transfection with 10 µg pCas-*Ov*-GRN-1 plasmid DNA in ~500 µl RPMI-1640 (Sigma) by electroporation; all 20 flukes were included in the same cuvette during electroporation. The electroporation was performed in 4 mm cuvettes (Bio-Rad) with a single square wave pulse of 125 volts for 20 ms using a Gene Pulser Xcell (Bio-Rad) (*Papatpremsiri et al., 2015*; *Piratae et al., 2012*). Flukes were then washed several times with 150 mM NaCl and an additional five times with RPMI-1640 containing 2× Pen/Strep. Flukes were cultured in RPMI-1640 containing 2× Pen/Strep at 37°C in 5% CO$_2$ atmosphere (*Papatpremsiri et al., 2015*; *Piratae et al., 2012*). Two control groups were included: wild-type (WT) mature flukes and 'mock' control flukes which were exposed to identical electroporation conditions with RPMI-1640 and 1× Pen/Strep in the absence of plasmid DNA. The adult flukes were observed and collected after 1, 2, 3, 5, 7, 14 and 21 days of culture following pCas-*Ov*-GRN-1 transfection. RNA and protein were extracted from individual flukes and *Ov*-GRN-1 mRNA expression was assessed by RT-qPCR and *Ov*-GRN-1 protein expression was assessed by western blot. Mutations and/or insertions-deletions (INDELs) resulting from CRISPR/Cas were estimated by two discrete types of analysis: 1) by Illumina-based Next Generation Sequencing (NGS) (*Shah et al., 2015*; *Albadri et al., 2017*); 2) by CRISPR efficiency estimation (*Shah et al., 2015*; *Yu et al., 2014*; *Yang et al., 2017*), a method based on the differences in RT-qPCR efficiencies between amplification of the WT and mutant sequence with a primer spanning the targeted mutation site.

MC and NEJ (750 larvae per cuvette) were subjected to square wave electroporation in the presence of pCas-*Ov*-GRN-1 pDNA as described above for adult flukes. The larvae were washed as above and cultured in RPMI complete medium (2× Pen/Strep) at 37°C in 5% CO$_2$ atmosphere. Transcript levels for *Ov*-GRN-1 on days 1, 2, 3, and 5 after transfection were ascertained by RT-qPCR, as above.

## Extraction of nucleic acids

RNA was extracted from pooled or individual transfected flukes using the TRIzol reagent (Invitrogen). Concentration of RNA was estimated by absorbance at 260 nm using a NanoVue spectrophotometer. Genomic DNA was extracted from individual adult flukes using the QIAamp DNA Mini Kit

(Qiagen). A dual RNA and DNA extraction was used for individual worms at day 60 after infection of hamsters with *Ov*-GRN-1 gene-edited NEJ, using RNAzol RT and DNAzol (Molecular Research Center, Inc.) (*Chan et al., 2014*; *Chen et al., 2010*). In brief, each worm was homogenized in RNAzol RT using a motorized pestle, the DNA and protein from the lysate was precipitated using DNAse-RNAse-free water. The aqueous phase (top) was transferred into isopropanol to precipitate the RNA. The DNA/protein pellet was resuspended in DNAzol, and DNA extracted as per the manufacturer's instructions. Expression levels of *Ov*-GRN-1 in total RNA recovered from individual liver flukes were determined.

To assess the performance of the gene editing approach, following necropsy of hamsters and recovery of the liver flukes, the adult worms were assigned to one of three phenotypes based on the levels of *Ov*-GRN-1 transcript knockdown, low (L), moderate (M) or high (H), as follows: L, ≥100% relative to WT mean (low efficiency of programmed genome editing), group termed $_L\Delta Ov$-GRN-1; M, >10 to<100% relative to WT mean, group termed $_M\Delta Ov$-GRN-1; and H, ≤10% relative to WT mean, group termed $_H\Delta Ov$-GRN-1. Pools of genomic DNAs from flukes, which had been assigned to each of the L, M and H groups of *Ov*-GRN1 transcript knockdown levels, were quantified for efficiency of CRISPR/Cas9-programmed gene editing in terms of mutation levels by qPCR and Illumina-based deep sequencing (below) (*Yang et al., 2017*; *Vasquez et al., 2018*). The data for the pooled samples from each group are based on a single Illumina run, that is n = 1 sample for each of the L, M and H genomic DNA pools.

## Quantitative real-time PCR

Complementary DNA (cDNA) was synthesized from parasite total RNA using an iScript cDNA synthesis kit (Thermo Fisher Scientific) prior to proceeding with reverse transcription quantitative real-time PCR (RT-qPCR). RT-qPCR was performed with biological triplicate samples using a SYBR Green kit (Takara Bio USA, Inc., Mountain View, CA) in a thermal cycler (Light Cycler 480 II, Roche Diagnostics GmbH, Mannheim, Germany). Each RT-qPCR reaction consisted of 7.5 μl SYBR Green Master Mix, 0.5 μl (10 μM) each of specific forward and reverse primers for *Ov*-GRN-1 (*Figure 1B*) (forward primer, *Ov*-GRN-1-RT-F: 5'-GGGATCGGTTAGTCTAATCTCC and reverse primer, *Ov*-GRN1-RT-R: 5'-GATCATGGGGGTTCACTGTC), amplifying 359 base pairs (bp) of the product (nt 7365 of *O. viverrini* granulin-1 mRNA, GenBank FJ436341.1), 2 μl of cDNA and distilled water to a final volume of 15 μl. The thermal cycle was a single initiation cycle at 95°C for 3 min followed by 40 cycles of denaturation at 95°C for 30 s, annealing at 55°C for 30s, extension at 72°C for 45s and a final extension at 72°C for 10 min. The endogenous actin gene (GenBank EL620339.1) was used as a housekeeping control (*Papatpremsiri et al., 2015*; *Piratae et al., 2012*; *Chaiyadet et al., 2017*) (forward primer, *Ov*-actin-F: 5'-AGCCAACCGAGAGAAGATGA and reverse primer *Ov*-actin-R: 5'-ACCTGACCATCAGGCAGTTC). The fold change in *Ov*-GRN-1 transcripts was calculated by the $2^{(-\Delta\Delta Ct)}$ method using *Ov*-actin for normalization (*Papatpremsiri et al., 2015*; *Piratae et al., 2012*; *Schmittgen and Livak, 2008*). Means and standard deviations were calculated and means compared by two-way ANOVA using GraphPad Prism software.

## Rabbit anti-*Ov*-GRN-1 antiserum and western blot analysis

One milligram of adjuvanted, recombinant *Ov*-GRN1 protein (*Smout et al., 2009*; *Strannegård and Yurchision, 1969*) was subcutaneously injected into an outbred New Zealand White rabbit. The rabbit was boosted twice with 500 μg of adjuvanted protein, and 2 weeks after the last booster the rabbit was euthanized after which blood was collected by cardiac puncture (Animal Ethics Committee, Khon Kaen University, approval no ACUC-KKU-61/60; see above). *Ov*-GRN-1 protein levels were determined by western blot using rabbit anti-recombinant *Ov*-GRN-1 antiserum. The adult flukes from either WT or Δ*Ov*-GRN-1 groups were collected individually at days 1, 2, 3, 5, 7, 14 and 21 after electroporation (three flukes per group). Groups of three flukes were homogenized by sonication (Sonics and Materials) in 1× PBS with alternating pulses of 5s duration (with 5s pause between pulses) for 45 s at 4°C. The homogenate was clarified by centrifugation at 13,000 ×*g* for 30 min at 4°C, after which the supernatant was stored at −20°C. Protein concentration of fluke homogenates was determined by the Bradford assay. Homogenates were subjected to SDS-PAGE (15%) electrophoresis, and the resolved proteins transblotted to nitrocellulose membrane using a Mini Trans-Blot Cell (Bio-Rad). Membrane strips containing 2 μg of total protein were washed with 0.5% Tween-20

in 1× PBS (PBST), blocked with 5% skimmed milk in PBST for 60 min and probed with rabbit anti-*Ov*-GRN-1 serum or pre-immunization serum, diluted 1:50 with 1% skimmed milk in PBST, for 2 hr with gentle agitation. After washing, the strips were probed with horseradish peroxidase (HRP)-goat anti-rabbit IgG (Invitrogen), diluted 1:1000 in antibody buffer, for 60 min. The strips were washed, signals detected using enhanced chemiluminescence (ECL) substrate (GE Healthcare Life Sciences) and imaged using an Image Quant LAS 4000 mini (GE Healthcare Life Sciences). As a control protein also derived from the tegument of *O. viverrini* flukes, we also assessed the protein expression levels of *Ov*-TSP-2 by western blot using a specific antibody raised to the recombinant protein (*Chaiyadet et al., 2017*). Relative protein expression levels as established by western blot were measured by densitometry using Image J, https://imagej.nih.gov/ij/download.html. Levels of protein expressed between groups were compared by independent Student's *t*-tests.

## CRISPR/Cas efficiency and mutation levels estimated by quantitative PCR

Adult flukes were collected on days 1, 2, 3, 5, 7, 14 and 21 after pCas-*Ov*-GRN-1 transfection. The genome of each fluke was investigated for mutation(s) expected to have resulted from the repair by NHEJ events following the sgDNA programmed double stranded break (DSB) of the *Ov*-GRN-1 locus by Cas9. For analysis of gDNA from individual adult liver flukes recovered from infected hamsters, we performed a qPCR assay to detect and quantify the frequencies of newly induced mutations. The approach employed two pairs of primers for the target locus, with one putative amplicon extending beyond the putative INDEL site and the other overlapping it, as described (*Yu et al., 2014*). The primers were named *Ov*-GRN-1-OUT-F, *Ov*-GRN-1-OVR-F, and *Ov*-GRN-1-reverse (OUT/OVR-R), respectively. The primer pair of *Ov*-GRN-1-OUT-F (5'-TTCGAGATTCGGTCAGCCG) and OUT/OVR-R (5'-TTGGTCGGCCAGTATGTTCG) amplified the fragment flanking and spanning the DSB (1,496–2,312 nt), whereas the primer pair *Ov*-GRN-1-OVR-F (5'-CAAGTGTTGACGGTGA TTTCACTT) and OUT/OVR-R amplified a region overlapping the DSB (1599–2312) (*Figure 1B*). Whereas both primer pairs exhibited equivalent amplification efficiencies with the genomic DNA template from WT flukes, the *Ov*-GRN-1-OVR-F and OUT/OVR-R primer pair was mutation sensitive, whereas the other pair was not. The OUT and OVR amplicons were 817 and 714 bp in size, respectively, using the following PCR conditions: 7.5 µl of SYBR Green Master Mix (TaKaRa Perfect Real-time Kit), 0.5 µl (0.4 µM) of each primer, 10 ng/µl of gDNA and distilled water to 15 µl. The thermal cycles included initiation for one cycle at 95℃, 3 min followed by 40 cycles of denaturation at 95℃, 30s, annealing at 55℃, 30s, extension at 72℃, 45s, and a final extension at 72℃ for 10 min. The SYBR green signal was read at each annealing cycle and reported as threshold cycle (Ct). Efficiency of programmed CRISPR/Cas editing was estimated as the ratio of $Ct_{OUT}:Ct_{OVR}$ from the experimental group compared with $Ct_{OUT}:Ct_{OVR}$ of the control group, as described (*Yu et al., 2014*). The $Ct_{OUT}:Ct_{OVR}$ ratio from the control group would equal '1' (CRISPR efficiency = 0) since there was difference in Ct values from the OUT and OVR primers. By contrast, the OVR primer can be anticipated to be inefficient when compared to the OUT primer for the experimental group, and hence the $Ct_{OUT}:Ct_{OVR}$ likely would be <1. Here, we calculated percent mutation indirectly by subtraction of the CRISPR/Cas9 efficiency value from '1', as indicated (*Yu et al., 2014*; *Sentmanat et al., 2018*).

$$\text{Efficiency}(F) = \frac{\text{Average}\,ct_{OUT}}{\text{Average}\,ct_{OVR}}$$

$$CRISPR/Cas9\,\text{efficiency} = \frac{F_{\Delta Ov-gm-1}}{F_{control}} \times (100)$$

$$\text{Mutation rate} = 100\% - CRISPR/Cas9\,\text{efficiency}$$

Genomic DNAs from flukes recovered from hamsters 60 days after infection with CRISPR/Cas9-treated NEJ, and which had been assigned to the low (L), moderate (M) or high (H) groups based on knockdown levels of *Ov*-GRN-1 transcripts, were pooled by group. The L, M and H groups were assessed and scored for efficiency of CRISPR/Cas9-programmed gene editing in terms of mutation levels by qPCR, as described above.

## Targeted Amplicon libraries, Illumina-based sequencing

Several Illumina NGS libraries were constructed. First, for analysis of programmed editing of adult flukes that were subjected to gene editing manipulation and subsequently cultured *in vitro*, genomic DNAs were extracted from the *Ov*-GRN-1 gene-edited adult liver flukes at each of 7, 14 and 21 days after transfection. A pool of gDNA was prepared from 15 of these flukes, from five worms from each time point. Second, gDNAs were pooled from 7 to 10 worms from each of the L, M, and H groups of ΔOv-GRN-1 worms (25 worms in total) (*Figure 5A and B*) and also a gDNA pool from 25 control non-gene-edited WT worms. Targeted amplicon NGS libraries were constructed from each of these of gDNA pools. In each case, an amplicon of 173 bp in size that spanned the DSB was amplified using *Ov*-GRN-1 MiSeq-F primer 5′-TTCGAGATTCGGTCAGCCG (position 1496–1514 nt) and *Ov*-GRN-1 MiSeq-R primer 5′-GCACCAACTCGCAACTTACA (position 1649–1668 nt) (*Figure 1B*). These amplicons were purified (Agencourt AMPure XP beads, Beckman) and ligated with Gene Read Adaptors Set A (Qiagen) and Illumina compatible adaptor(s) and barcode(s) using QIAseq 1-step Amplicon library kit (Qiagen). The libraries were quantified using the GeneRead Library Quant Kit (Qiagen) with Illumina index/barcode specific primers, and concentration of the libraries established using standard libraries provided in the kit. Illumina NGS was performed by GENEWIZ (South Plainland, NJ). Index/adaptor and primer out sequences were trimmed from the reads. Analysis of the sequenced reads using the SnapGene (GSL Biotech LLC) and the CRISPResso software (https://github.com/lucapinello/CRISPResso) suites was carried out to validate and characterize programmed mutations of the alleles, including assessment of NHEJ-induced INDELS as insertions, deletions and/or substitutions (*Canver et al., 2018*; *Pinello et al., 2016*). The sequences of the alleles were compared to the reference sequence represented by the target amplicon of the WT *Ov*-GRN-1 gene (GenBank FJ436341.1) and to the reads from the control worms for the flukes derived from infection of hamsters with gene-edited NEJ. Of these two analysis methods for performance of CRISPR/Cas9 gene-editing, the qPCR approach (*Yu et al., 2014*) is quick and inexpensive in comparison to the targeted amplicon NGS approach (*Canver et al., 2018*; *Shalem et al., 2015*). However, the latter approach provides more detailed characterization of the events including the types and frequencies of the INDELS, and is more accurate (*Sentmanat et al., 2018*).

## Cell proliferation and wound healing assays

To evaluate the effect of *Ov*-GRN-1 gene editing on liver fluke-driven proliferation of human cholangiocytes, motile WT or ΔOv-GRN-1 adult flukes were co-cultured with cells of the human cholangiocyte cell line H69 in 24-well Trans-well plates (three wells per group) (*Papatpremsiri et al., 2015*) containing a 4 µm pore size membrane separating the upper and lower chambers (Corning). In brief, 15,000 H69 cells were seeded into the lower chamber of the plate and cultured with complete medium containing DMEM/F12 supplemented with $1 \times$ antibiotic, 10% fetal bovine serum, 25 µg/ml adenine, 5 µg/ml insulin, 1 µg/ml epinephrine, 8.3 µg/ml holo-transferrin, 0.62 µg/ml hydrocortisone, 1.36 µg/ml T3, and 10 ng/ml epidermal growth factor (*Ninlawan et al., 2010*) for 24 hr, after which the cells were fasted for 4–6 hr in medium supplemented with only one twentieth of the growth factor content of complete medium. Five viable *O. viverrini* adult flukes that had been transfected (or not) with pCas-*Ov*-GRN-1 pDNA in a total of 500 µl of RPMI (or medium alone) were placed into the upper chamber of each well. The number of cells in each well was determined at days 1, 2, and 3 using $1 \times$ PrestoBlue cell viability reagent (Invitrogen) (*Tynan et al., 2012*) added to cells at 37°C in 5% $CO_2$ atmosphere for up to 60 min. Cell number was determined at 570 nm and calculated from a standard curve before transforming into relative growth compared to control groups. Cell proliferation assays were carried out in triplicate.

To assess the effect of *Ov*-GRN-1 knockout on *in vitro* wound healing, 300,000 cholangiocytes in monolayers were grown in 6-well Trans-well plates with a 4 µm pore size. These cells were cultured in complete media for 2 days at 37°C in 5% $CO_2$ atmosphere then transferred to incomplete media overnight. Monolayers in each well were scratched using a sterile 200 µl autopipette tip (*Papatpremsiri et al., 2015*; *Smout et al., 2015*; *Liang et al., 2007*) and washed with 1× PBS twice to remove disconnected cells or debris. Ten transfected adult or control flukes were added to the upper chamber of the Transwell plate containing the wounded cell monolayer in the lower chamber. The rate of wound closure was measured at 0, 12, 24 and 36 hr, respectively. Transwell plates were imaged using an inverted microscope (Nikon) and images of all groups were captured at all-time

points quantitatively using Adobe Photoshop CS6. The distances between different sides of the cell monolayer scratch were measured by drawing a line in the middle of the scratch on the captured image (*Papapremsiri et al., 2015*; *Smout et al., 2015*; *Liang et al., 2007*; *Smout et al., 2011*). The analysis of monolayer wound healing was repeated three times.

H69 cells (*Smout et al., 2015*; *Grubman et al., 1994*) were authenticated using STR profiling by PCR by ATCC and were confirmed in our laboratory to be *Mycoplasma*-free using the Lookout Mycoplasma PCR detection kit (Sigma-Aldrich).

## Infection of hamsters with *Ov*-GRN-1 gene-edited NEJs and histopathological assessment of hepatobiliary lesions

Thirty male Syrian golden hamsters, 6–8 weeks of age, were obtained from the Animal Unit, Faculty of Medicine, Khon Kaen University (approval number ACUC-KKU-61/60). The hamsters were randomly divided into three groups of 10 animals per group: uninfected control, infected with WT flukes, and infected with ΔOv-GRN-1 flukes. Each hamster was infected with 100 active NEJs through intragastric intubation; the uninfected control group was fed normal saline solution instead of NEJ (*Sripa and Kaewkes, 2002*). Hamsters (five animals per cage) were maintained under conventional conditions and fed a stock diet (C.P. Ltd., Thailand) and water *ad libitum* until they were euthanized (*Sripa and Kaewkes, 2002*). Following euthanasia, five hamsters from each group were necropsied for histopathological assessment of the hepatobiliary tract at days 14 and 60 post-infection (*Sripa and Kaewkes, 2002*). The hamsters were euthanized by overdose of anesthesia with diethyl ether. Subsequently, blood was obtained by cardiac puncture and the livers were removed. Fluke numbers were counted from two livers of both WT and ΔOv-GRN-1 groups at day 60 post-infection and compared using an unpaired two-tailed *t*-test. The left and right lobes of the liver from five hamsters were dissected, cross-sectioned, and each lobe was divided into three parts. The liver fragments were fixed in 10% buffered formalin and stored overnight at 4°C before processing. Formalin-fixed liver was dehydrated through an ethanol series (70, 95, and 100%), cleared in xylene, and embedded in paraffin. Paraffin embedded sections of 4 μm thickness, cut by microtome, were stained with hematoxylin and eosin (H&E) or Picro-Sirius Red, or probed with anti-ACTA2 antibodies, and analyzed for pathologic changes (below).

## Biliary hyperplasia

H&E staining was used to assess pathological changes. The sections were deparaffinized in 100% xylene, rehydrated through a descending series of alcohol, stained with H&E for 5 min, dehydrated in an ascending series of alcohol, cleared with 100% xylene, mounted in Permount medium on a glass slide, and slides were dried overnight at 37°C and photographed under light microscopy. Images (200×) from H&E-stained sections from five hamsters infected with WT flukes, five hamsters infected with ΔOv-GRN-1 flukes, and five uninfected hamsters were assessed. Thickness (width) of the bile duct epithelium from each thin liver section was measured with ImageJ at eight equidistant positions around the bile duct. To compensate for outliers, the median width for each bile duct was used for the analysis. The two-way ANOVA Holm-Sidak multiple comparisons test was used to compare groups at each time point.

## Fibrosis

Two stains were used separately to assess biliary fibrosis. First, sections were stained with Picro-Sirius Red (Abcam, Cambridge Science Park, UK). Sufficient Picro-Sirius Red solution was applied to completely cover the tissue sections on the slide, the stained slide was incubated at ambient temperature for 60 min, rinsed in two changes of acetic acid solution and dehydrated through two changes of absolute ethanol. Slides were cleared with 100% xylene, mounted in Per-mount, dried overnight at 37°C and photographed by light microscopy to document collagen surrounding the bile ducts. ImageJ was used to auto-color balance the images using the macro by Vytas Bindokas at https://digital.bsd.uchicago.edu/docs/imagej_macros/_graybalancetoROI.txt followed by application of the MRI fibrosis tool to quantify percentage area of fibrosis (red-stain) at default settings (red 1: 0.148, green 1: 0.772, blue 1: 0.618, red 2: 0.462, green 2: 0.602, blue 2: 0.651, red 3: 0.187, green 3: 0.523, blue 3: 0.831) (*Pereira, 2016*). Twenty discrete images (200×) stained with Picro-Sirius Red from each hamster (five animals per treatment group) were assessed, that is 100 images per group.

Given a broad range of values among groups, comparison of the groups was undertaken using the Kruskal-Wallis with Dunn's multiple comparisons test.

Expression level of smooth muscle alpha-actin ($\alpha$-SMA; ACTA2) also was assessed as a surrogate for fibrosis. Liver sections from hamsters at 60 days post-infection were deparaffinized 3 times with 100% xylene, 5 min each. Sections were rehydrated with an ascending series of ethanol; 100%, three times, 3 min each, 95%, three times, 3 min each, 70% for 3 min, followed by thorough washing in tap water for 5 min, distilled water for 5 min, and $1\times$ PBS for 5 min. Thereafter, slides were incubated in citrate buffer 0.1 M, pH 6.0 (citric acid, anhydrous, 0.06 M, sodium citrate dihydrate, 0.04 M) at 110°C for 5 min, allowed to cool at room temperature for 20 min, and then washed in $1\times$ PBS, three times, 5 min each. Thereafter, the sections were blocked with 5% bovine serum albumin (BSA) for 30 min in a humidified chamber and washed in three times in $1\times$ PBS, 3 min each with occasional shaking. The slides were probed with Alexa Fluor 594-labeled anti-ACTA2 antibody (Abcam) diluted 1:200 in 1% BSA in PBST, 18 hr at 4°C in a humidified atmosphere. Lastly, slides were washed as above, mounted in glycerol, diluted 1:4 with $1\times$ PBS, and examined under bright and fluorescent lights (Zeiss Axio Observer; AxioVision SE64 Rel. 4.9.1 software, Jena, Germany). Images with a bile duct containing a fluke were selected and ImageJ used to define regions adjacent to the epithelium that excluded potential blood vessels (ovate structures). Three sites, free of bile ducts and blood vessels, were selected at random in order to establish levels of background fluorescence, each comprising 5–10% of the image. The fluorescence intensity of the biliary epithelium was measured and blanked against the mean of the three background readings and reported as mean intensity per $cm^2$ at 300 pixels per inch (PPI). Twenty-five to 30 discrete images of bile ducts per treatment group (three hamsters) were assessed. Zero values from the uninfected group were assigned a value of 1 to enable use of a log axis. The groups were compared using one-way ANOVA with Holm-Sidak multiple comparisons test.

## Biological and technical replicates, statistics

These biological replicates represented parallel measurements of biologically discrete samples in order to capture any random biological variation. Technical replicates were undertaken as well; these represented repeated measurements of the same sample undertaken as independent measurements of the random noise associated with the investigator, equipment or protocol.

Means for experimental groups were compared to control by one or two ways ANOVA and where appropriate, by Student's $t$-test (GraphPad Prism, La Jolla, CA). Values for p of $\leq 0.05$ were considered to be statistically significant.

## Acknowledgements

We thank Suwit Balthaisong and Meredith Brindley for technical assistance and scientific illustrations, and the collaboration of colleagues of the Tomsk OPIsthorchiasis Consortium, TOPIC, www.topic-global.org. This study was supported by the Thailand Research Fund through the Royal Golden Jubilee PhD Program (grant no. PHD/0111/2557 to PA and TL), the National Cancer Institute (NCI), US National Institutes of Health (NIH) award R01CA164719 (AL, TL, PJB,) and the National Health and Medical Research Council awards APP1085309 (AL, JS, TL), senior principal research fellowship APP1117504 (AL), and career development fellowship award APP1109829 (NDY). These studies were supported in part by Wellcome Trust (WT) Strategic Award number 107475/Z/15/Z (Karl F Hoffmann, principal investigator; PJB, co-investigator). The content is solely the responsibility of the authors and does not necessarily represent the official views of the NCI, NIH, NHMRC or WT.

## Additional information

### Funding

| Funder | Grant reference number | Author |
| --- | --- | --- |
| Thailand Research Fund | PHD/0111/2557 | Patpicha Arunsan Thewarach Laha |
| National Health and Medical Research Council | APP1109829 | Neil David Young |

| National Health and Medical Research Council | APP1085309 | Alex Loukas<br>Javier Sotillo<br>Thewarach Laha |
| --- | --- | --- |
| National Cancer Institute | R01CA164719 | Alex Loukas<br>Thewarach Laha<br>Paul J Brindley |
| National Health and Medical Research Council | APP1117504 | Alex Loukas |
| Wellcome | 107475/Z/15/Z | Paul J Brindley |

The funders had no role in study design, data collection and interpretation, or the decision to submit the work for publication.

## Author contributions

Patpicha Arunsan, Resources, Formal analysis, Funding acquisition, Validation, Investigation, Visualization, Methodology, Writing—original draft, Writing—review and editing, Contributed equally with Wannaporn Ittiprasert and Michael J. Smout; Wannaporn Ittiprasert, Conceptualization, Data curation, Software, Formal analysis, Supervision, Validation, Investigation, Visualization, Methodology, Writing—original draft, Writing—review and editing; Michael J Smout, Software, Formal analysis, Validation, Investigation, Visualization, Methodology, Writing—original draft, Writing—review and editing, Contributed equally with Patpicha Arunsan and Wannaporn Ittiprasert; Christina J Cochran, Conceptualization, Formal analysis, Investigation, Methodology; Victoria H Mann, Resources, Investigation, Visualization, Project administration, Writing—review and editing; Sujittra Chaiyadet, Shannon E Karinshak, Investigation, Visualization; Banchob Sripa, Supervision, Methodology, Writing—review and editing; Neil David Young, Conceptualization, Resources, Investigation; Javier Sotillo, Investigation, Methodology, Writing—review and editing; Alex Loukas, Conceptualization, Supervision, Funding acquisition, Validation, Writing—original draft, Writing—review and editing; Paul J Brindley, Conceptualization, Resources, Data curation, Formal analysis, Supervision, Funding acquisition, Investigation, Methodology, Writing—original draft, Project administration, Writing—review and editing; Thewarach Laha, Conceptualization, Resources, Formal analysis, Supervision, Funding acquisition, Validation, Investigation, Methodology, Writing—original draft, Project administration, Writing—review and editing

## Author ORCIDs

Wannaporn Ittiprasert  http://orcid.org/0000-0001-9411-8883
Michael J Smout  https://orcid.org/0000-0001-6937-0112
Shannon E Karinshak  https://orcid.org/0000-0002-2079-0973
Neil David Young  https://orcid.org/0000-0001-8756-229X
Javier Sotillo  https://orcid.org/0000-0002-1443-7233
Paul J Brindley  https://orcid.org/0000-0003-1765-0002

## Ethics

Animal experimentation: The Animal Ethics Committee of Khon Kaen University approved the study, approval number ACUC-KKU-61/60, which adhered to standard guidelines of the National Research Council of Thailand for the Ethics of Animal Experimentation.

## Decision letter and Author response

Decision letter https://doi.org/10.7554/eLife.41463.018
Author response https://doi.org/10.7554/eLife.41463.019

# Additional files

## Supplementary files

• Transparent reporting form
DOI: https://doi.org/10.7554/eLife.41463.012

## Data availability

The Illumina sequencing reads and related project details are available at GenBank: Bioproject PRJNA385864, Biosample SAMN07287348, SRA study SRP110673, accessions SRR5764463-5764618 and SRR8187484-SRR8187487, at https://www.ncbi.nlm.nih.gov/Traces/study/?acc=SRP110673, Bioproject, www.ncbi.nlm.nih.gov/bioproject/PRJNA385864.

The following dataset was generated:

| Author(s) | Year | Dataset title | Dataset URL | Database and Identifier |
|---|---|---|---|---|
| Wannaporn Ittiprasert | 2017 | *Opisthorchis viverrini* Raw sequence reads | https://www.ncbi.nlm. nih.gov/bioproject/ 385864 | GenBank, PRJNA385864 |

The following previously published dataset was used:

| Author(s) | Year | Dataset title | Dataset URL | Database and Identifier |
|---|---|---|---|---|
| Smout MJ, Mulvenna J, Laha T, Sripa B, Brindley PJ, Loukas A | 2009 | Sequence: FJ436341.1 | https://www.ebi.ac.uk/ ena/data/view/FJ436341 | European Nucleotide Archive, FJ436341 |

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
