## [Decision Letter]

Thank you for submitting your article "Programmed knockout mutation of liver fluke granulin attenuates virulence of infection-induced hepatobiliary morbidity" for consideration by *eLife*. Your article has been reviewed by three peer reviewers, including James B Lok as the Reviewing Editor and Reviewer #3, and the evaluation has been overseen by Wendy Garrett as the Senior Editor.

The reviewers have discussed the reviews with one another and the Reviewing Editor has drafted this decision to help you prepare a revised submission.

Summary:

All three reviewers concur that this paper represents a ground breaking advancement in functional genomic methodology for parasitic flatworms, and we congratulate you for this. We also concur that your findings are of special importance in that they demonstrate the utility of CRISPR/Cas9 mutagenesis in elucidating pathogenic mechanisms in *O. viverrini* and parasitic flatworms generally. All reviewers concur that the findings reported in this paper are sufficient to warrant publication in *eLife* and that no further experimentation is required. That said, there are several substantive points that should be addressed, given that this paper, along with its co-submitted manuscript on *S. mansoni* omega 1, will likely be prototypes upon which further studies involving programmed gene editing will be based. The general areas in need of this attention are 1) the difficulty in reconciling the very low mutation frequencies determined by deep sequencing with the very large decrements seen in parameters of target gene expression (mRNA, protein) and parasite virulence assessed on a worm-for-worm basis, 2) the need for clarification of how pooling of parasites and assignment to categories of high, medium and low were carried out and 3) addressing the rigor of controls used in the experiments. These concerns may be addressed by clarifying the points indicated and adding in-depth discussion of what could be improved with regard to informative indices of mutation frequency and proper controls going forward.

Essential revisions:

With regard to point 1 in the summary, provide an assessment of the deep sequencing approach as an informative estimate of mutation frequency compared with possible alternative methods that would yield an estimation of the frequency of mutations in the target on a worm-for-worm basis. Comments on how the low mutation frequencies obtained by the deep sequencing approach can be reconciled with the dramatic decrements in *Ov-grn-1* transcripts and protein and in the virulence of the parasites would also greatly improve the paper. As suggested by reviewer 1 (and by all of us in consultation) a discussion of the site of *Ov-grn-1* expression within the worms as it relates to efficiency of transduction with CRISPR elements would also be helpful.

With regard to point 2 in the summary, the segment on "Longevity of programmed mutation at *Ov-grn-1*" was somewhat confusing to all three reviewers before and during our consultation. It is essential to clarify this section along lines specified by reviewers 1 and 3.

With regard to point 3 in the summary, the three reviewers concurred that a control in which Cas9 was expressed without a functional gRNA would have added to the rigor of the study. There was difference of opinion about the advisability of including in such a control a nonfunctional gRNA (i.e. one that contained a scaffold with no seed sequence or that contained a scaffold with a seed sequence having no homology in the *O. viverrini* genome). All this is to say that since this paper is likely to constitute a prototype for future studies, some discussion of what constitutes a rigorous control in experiments of this kind is warranted.

*Reviewer #1:*

This manuscript describes the first report of CRISPR driven gene editing in a parasitic flatworm, in this case the oriental liver fluke *Opisthorchis viverrini*. The focus of the transfection efforts is liver fluke granulin, shown previously to play a role in establishing the tumorigenic microenvironment.

The authors effectively demonstrate successful transfection and then provide extensive analysis of the model host pathology induced by transfected fluke relative to wild-type (control) fluke. The manuscript charts an important step forward in the field, providing evidence for the utility of CRISPR/Cas9 in flatworms. The work on granulin biology extends the current understanding of its role. Overall, the manuscript is generally well presented and the results are compelling.

The authors should address the following points for clarification etc.

It would have been helpful to discuss where granulin is expressed in the worm as this can help inform the tissue-specific efficiency of the transfection methods used.

The efficiency of transfection data was provided for the directly transfected adults and yet the biology was performed on adults derived from transfected of NEJs – it's unclear why the transfection data were not generated for the juvenile flukes OR from adults of transfected NEJs.

Mutation frequencies are provided for adult fluke with low, medium and high levels of target gene transcript expression. It's not clear if the data for the pooled samples from each group are based on an n=1 or multiple pools of each type. Either way, this needs to be clarified and, if multiple pools were used, the range or SE values included with all the relevant data in the manuscript..

The ranges used to assign to define the H, M and L groups appear to differ between the Results (subsection “Longevity of programmed mutation at *Ov-grn-1*”) and Materials and methods (subsection “Extraction of nucleic acids”).

In the subsection “Programmed mutation of growth factor secreted by carcinogenic liver fluke”, the three outcomes of transfection (insertion, deletion, substitution) totaled 27640 against the total number of NHEJ reads of 27616 – but why these differ is not stated.

Subsection “Attenuated infection-induced hyperplasia of the biliary tract”, last – sentence is incomplete and needs rewriting.

Materials and methods – Wrt transfection of mature adult fluke, the authors need to state if the 20 worms were transfected individually (1 per cuvette) or otherwise?

*Reviewer #2:*

The paper of Arunsan et al. reports on the functional characterization of *O. viverini* granulin, a secreted parasite protein. By CRISPR/Cas9-mediated mutagenesis and a combination of molecular analyses, cell culture and animal experiments as well as histochemical studies, the authors demonstrate a role of granulin in the liver/bile duct-associated pathology in the hamster as a host model.

This is an elegant and well-designed study showing for the first time that CRISPR/Cas9-mediated KO studies are possible in an important human parasite. Along with the accompanying paper about schistosome CRISPR/Cas9-mediated KO, this is groundbreaking work and worth to be published in *eLife*.

*Reviewer #3:*

This manuscript confirms the overall role of the liver fluke *Opisthorchis viverrini*, and its secreted granulin 1 as direct agents of hepatobiliary disease, including cholangiosarcoma, in humans. But beyond these confirmatory findings, this paper is groundbreaking in that its evidence derives from phenotypes resulting from CRISPR/Cas9-programmed disruption of *Ov-grn-1*, the gene encoding the parasite's pathogenic secreted granulin-1 protein. This, in combination with its co-submitted manuscript on CRISPR/Cas9 disruption of the gene encoding the secreted egg protein, omega-1 in *Schistosoma mansoni*, at once marks proof of principle for programmed gene editing in pathogenic trematodes and the first experiments in which this powerful method is applied to determining pathogenicity of specific molecules in these important parasites. In this paper, the authors demonstrate by deep sequencing the creation of indels at the target locus in the *Ov-grn-1* gene and estimate frequency of these among PCR amplicons of the target locus in from pooled parasites. They go on to demonstrate that these mutations correlate with marked declines in *Ov-grn-1* transcript and protein levels as well as decreases in the capacity of parasite secretions to incite cell proliferation and simulated wound healing in vitro. Finally, they report that although parasites with CRISPR-induced disruption of *Ov-grn-1* retain the ability to infect hamsters, the levels of thickening and structural disorder they incite in biliary epithelia of these animals and the fibrosis resulting from this is greatly reduced from those induced by wild type parasites. This is a highly significant milestone in the study of molecular determinants of pathogenicity in trematode parasites. That said, I noted two substantive points about the manuscript that, if not addressed by additional experiments in the present instance, should at least be acknowledged to improve upon this approach in future studies.

Substantive issues:

1) If I understand the narrative under "Longevity of programmed mutation at *Ov-grn-1*" it suggests that individual flukes were subjected to quantitative RT-PCR to determine levels of expression, classed according to three levels of expression: low, medium and high, and then their gDNA pooled for deep sequencing to determine mutation rates. Is it not possible to genotype individual flukes, or does mosaicism tend to render mutant genomes in these worms practically undetectable?

If possible to obtain, an estimate of the percentage of worms in a sample population that showed any mutation might be easier to reconcile with the very large reductions you see in *Ov-grn-1* protein and the large changes in the parameters relating to hepatobiliary disease that you see.

2) The control groups (subsection “Transfection of liver flukes with pCas-*Ov-grn-1”*) in this study appear to have been WT parasites and "mock transected" parasites that were electroporated in complete medium lacking plasmid DNA. Wouldn't a more rigorous control be parasites electroporated with a vector containing all functionalities except that the target-specific gRNA was replaced with either a gRNA with a scaffold but no seed sequence, or, better yet, a gRNA containing a seed sequence with no homology in the *O. viverrini* genome? This would control for any non-specific effects of Cas9 expression on parasite fitness. Such effects have been noted in other systems, notably in *C. elegans* lines that express Cas9 constitutively.

---

## [Author Response]

Essential revisions:With regard to point 1 in the summary, provide an assessment of the deep sequencing approach as an informative estimate of mutation frequency compared with possible alternative methods that would yield an estimation of the frequency of mutations in the target on a worm-for-worm basis. Comments on how the low mutation frequencies obtained by the deep sequencing approach can be reconciled with the dramatic decrements in Ov-grn-1 transcripts and protein and in the virulence of the parasites would also greatly improve the paper. As suggested by reviewer 1 (and by all of us in consultation) a discussion of the site of Ov-grn-1 expression within the worms as it relates to efficiency of transduction with CRISPR elements would also be helpful.

We have addressed the issue in the revised Discussion, second paragraph, as follows”

“Although the findings demonstrated programmed gene editing of the *Ov-grn-1* locus, the somatic mutation rate in the adult developmental stage was generally <5% of the genomes recovered from these multicellular parasites. […] If so, this may explain the marked reduction of expression and secretion of *Ov*-GRN-1 in tandem with a limited rate of mutation estimated in genomic DNA pooled from the gene-edited flukes.”

With regard to point 2 in the summary, the segment on "Longevity of programmed mutation at Ov-grn-1" was somewhat confusing to all three reviewers before and during our consultation. It is essential to clarify this section along lines specified by reviewers 1 and 3.

We revised the relevant sections in the Results and Materials and methods, extensively, including a revised heading in the Results and an additional figure (Figure 5) to provide the findings of a NGS/CRISPResso analysis of Illumina sequence reads of pools of genomic DNAs from the L, M and H groups in Figure 4. In the Materials and methods, we now include more thorough descriptions of the two approaches employed to quantify the outcome of programmed gene editing, 1) tri-primer qPCR and 2) targeted amplicon library NGS with analysis using the CRISPResso algorithm and software program; and to comparatively describe their attributes and limitations.

We anticipate that the revised text, including the new findings/Figure 5, will clarify the results and clarify our interpretation of the findings.

Results

“Gene editing efficiency negatively correlated with granulin expression during infection

Bile ducts parasitized by the gene-edited worms displayed a broad range of fibrosis from minimal to marked, as established by staining both with Sirius Red and with antibody specific for alpha-smooth muscle actin. […] Lastly, these findings also demonstrated the longevity of the programmed mutation at *Ov-grn-1*; mutations were retained in the parasite for at least 60 days during active infection of the mammalian host.”

Materials and methods

“Targeted amplicon libraries, Illumina-based sequencing

Several Illumina NGS libraries were constructed. First, for analysis of programmed editing of adult flukes that were subjected to gene editing manipulation and subsequently cultured in vitro, genomic DNAs were extracted from the *Ov-grn-1* gene-edited adult liver flukes at each of 7, 14 and 21 days after transfection. […] However, the latter approach provides more detailed characterization of the events including the types and frequencies of the INDELS, and is more accurate (Schmittgen and Livak, 2008).”

With regard to point 3 in the summary, the three reviewers concurred that a control in which Cas9 was expressed without a functional gRNA would have added to the rigor of the study. There was difference of opinion about the advisability of including in such a control a nonfunctional gRNA (i.e. one that contained a scaffold with no seed sequence or that contained a scaffold with a seed sequence having no homology in the O. viverrini genome). All this is to say that since this paper is likely to constitute a prototype for future studies, some discussion of what constitutes a rigorous control in experiments of this kind is warranted.

To address the concern, the revised Discussion includes a new paragraph addressing the reviewers/editors’ advice on additional controls including a non-functional gRNA:

Discussion

“The rigor of future gene editing investigations might be enhanced with the inclusion of additional controls including parasites transfected with an otherwise functional vector that lacks target-specific gRNA and/or a gRNA with a scaffold but without seed sequence and/or containing a seed sequence without homology in the genome of *O. viverrini*. […] Characterizing by immunolocalization the site of expression in the parasite from hamsters infected with gene-edited NEJ and/or the location of the gene editing plasmid after transfection of the liver fluke should be instructive.”

Reviewer #1:[…] It would have been helpful to discuss where granulin is expressed in the worm as this can help inform the tissue-specific efficiency of the transfection methods used.

We thank the reviewer for the useful suggestion – see response to point 1 above.

The efficiency of transfection data was provided for the directly transfected adults and yet the biology was performed on adults derived from transfected of NEJs – it's unclear why the transfection data were not generated for the juvenile flukes OR from adults of transfected NEJs.

Please see our response above to essential revision point 2.

Mutation frequencies are provided for adult fluke with low, medium and high levels of target gene transcript expression. It's not clear if the data for the pooled samples from each group are based on an n=1 or multiple pools of each type. Either way, this needs to be clarified and, if multiple pools were used, the range or SE values included with all the relevant data in the manuscript..

We have clarified this issue with the inclusion of the following statement in the Materials and methods section:

“The data for the pooled samples from each group are based on a single Illumina run, i.e. n = 1 sample for each of the L, M and H genomic DNA pools.”

The ranges used to assign to define the H, M and L groups appear to differ between the Results (subsection “Longevity of programmed mutation at Ov-grn-1”) and Materials and methods (subsection “Extraction of nucleic acids”).

Text revised to resolve the inconsistency, as follows:

Results section

“[…] adult flukes at necropsy were assigned to one of three groups based on *Ov-grn-1*mRNA expression levels, as follows: (i) ≥ 100% relative to WT mean, i.e., low (L) efficiency of programmed gene editing; group was termed _L_Δ*Ov-grn-1*; (ii) > 10 to < 100% relative to WT mean, i.e., moderate (M) level efficiency of programmed gene editing; termed _M_Δ*Ov-grn-1*; and (iii) ≤ 10% relative to WT mean, i.e., high (H) level efficiency of programmed gene editing; termed _H_Δ*Ov-grn-1*.”

Materials and methods section

”To assess the performance of the gene editing approach, following necropsy of hamsters and recovery of the liver flukes, the adult worms were assigned to one of three phenotypes based on the levels of *Ov-grn-1* transcript knockdown, low (L), moderate (M) or high (H), as follows: L, ≥100% relative to WT mean (low efficiency of programmed genome editing), group termed _L_Δ*Ov-grn-1*; M, > 10 to < 100% relative to WT mean, group termed _M_Δ*Ov-grn-1*; and H, ≤ 10% relative to WT mean, group termed _H_Δ*Ov-grn-1*.”

In the subsection “Programmed mutation of growth factor secreted by carcinogenic liver fluke”, the three outcomes of transfection (insertion, deletion, substitution) totaled 27640 against the total number of NHEJ reads of 27616 – but why these differ is not stated.

Text revised to resolve the inconsistency, as follows: “The CRISPResso pipeline was used to quantify gene-editing outcomes and efficiency (Canver et al., 2018; Pinello et al., 2016); among > 2 million reads aligned against the reference sequence, 27,640 sequence reads exhibited non-homologous end joining (NHEJ) mutations, including 170 reads with insertions (0.6%), 193 reads with deletions (0.7%) and 27,277 reads with substitutions (98.7%).”

Subsection “Attenuated infection-induced hyperplasia of the biliary tract”, last sentence is incomplete and needs rewriting.

Rewritten as:

“At 60 days after infection, significant differences in biliary hyperplasia remained between hamsters infected with WT (216%) and Δ*Ov-grn-1* (162%) flukes (*P* ≤ 0.05), although this was less marked than during acute infection at day 14 (Figure 3G).”

Materials and methods – Wrt transfection of mature adult fluke, the authors need to state if the 20 worms were transfected individually (1 per cuvette) or otherwise?

Clarified in the revised version, as follows:

“Pools of 20 mature adult flukes were simultaneously subjected to transfection with 10 µg pCas-*Ov-grn-1*plasmid DNA in ~500 µl RPMI-1640 (Σ) by electroporation; all 20 flukes were included in the same cuvette.”

Reviewer #3:1) If I understand the narrative under "Longevity of programmed mutation at Ov-grn-1" it suggests that individual flukes were subjected to quantitative RT-PCR to determine levels of expression, classed according to three levels of expression: low, medium and high, and then their gDNA pooled for deep sequencing to determine mutation rates. Is it not possible to genotype individual flukes, or does mosaicism tend to render mutant genomes in these worms practically undetectable?

First, please see our response above (essential revision point 2). Second, we did directly genotype individual flukes using the tri-primer qPCR approach (Figure 5). Second, it is technically feasible also to genotype individual worms using the targeted amplicon library-NGS-CRISPResso computational algorithm-based analysis, and whereas this approach will provide more information including the NHEJ profile of% insertions,% deletions, and% substitutions, it is technically more challenging, more time consuming, and more expensive.

If possible to obtain, an estimate of the percentage of worms in a sample population that showed any mutation might be easier to reconcile with the very large reductions you see in Ov-grn-1 protein and the large changes in the parameters relating to hepatobiliary disease that you see.

We have now addressed this issue experimentally following the reviewers’ recommendation; please see Figure 5 and related text. Also, we addressed the issue in our responses above to the essential revisions, points 1 and 2.

2) The control groups (subsection “Transfection of liver flukes with pCas-Ov-grn-1”) in this study appear to have been WT parasites and "mock transected" parasites that were electroporated in complete medium lacking plasmid DNA. Wouldn't a more rigorous control be parasites electroporated with a vector containing all functionalities except that the target-specific gRNA was replaced with either a gRNA with a scaffold but no seed sequence, or, better yet, a gRNA containing a seed sequence with no homology in the O. viverrini genome? This would control for any non-specific effects of Cas9 expression on parasite fitness. Such effects have been noted in other systems, notably in C. elegans lines that express Cas9 constitutively.

See our response to point 3 above.